# Deep and fast label-free Dynamic Organellar Mapping

Julia P. Schessner [1], Vincent Albrecht [1], Alexandra K. Davies [1,2], Pavel Sinitcyn [3] & Georg H. H. Borner [1] ✉

The Dynamic Organellar Maps (DOMs) approach combines cell fractionation and shotgun-proteomics for global profiling analysis of protein subcellular localization. Here, we enhance the performance of DOMs through data-independent acquisition (DIA) mass spectrometry. DIA-DOMs achieve twice the depth of our previous workflow in the same mass spectrometry runtime, and substantially improve profiling precision and reproducibility. We leverage this gain to establish flexible map formats scaling from high-throughput analyses to extra-deep coverage. Furthermore, we introduce DOM-ABC, a powerful and user-friendly open-source software tool for analyzing profiling data. We apply DIA-DOMs to capture subcellular localization changes in response to starvation and disruption of lysosomal pH in HeLa cells, which identifies a subset of Golgi proteins that cycle through endosomes. An imaging time-course reveals different cycling patterns and confirms the quantitative predictive power of our translocation analysis. DIA-DOMs offer a superior workflow for label-free spatial proteomics as a systematic phenotype discovery tool.

The compartments of eukaryotic cells organize the proteome into dynamic reaction spaces that control protein activity. The large number of diseases caused by disrupted protein transport demonstrates that protein localization must be tightly regulated to ensure correct protein function[1,2]. Our understanding of cellular homeostasis thus requires a comprehensive view of protein localizations and movements within the cell[3]. The growing field of spatial proteomics provides diverse approaches to study protein localization on the whole proteome scale[3–5]. Broadly, methods can be categorized into those based on high-throughput imaging[6,7], and those based on quantitative mass spectrometry (MS). The latter group includes methods that determine protein localization via protein interaction networks[8], as well as methods that determine protein localization more directly via organellar profiling[4].

Organellar profiling relies on the partial separation of organelles from cell lysates by fractionation, based on their differing physical properties (size/density). MS is then applied to quantify the relative abundance of proteins across the fractions, yielding protein abundance profiles (Supplementary Note 1) that are characteristic of the harbouring organelles. Unlike imaging- or interaction-based techniques, organellar profiling provides localization information for thousands of proteins in a single experiment. Established organellar profiling approaches include Protein Correlation Profiling (PCP)[9], Localization of organelle proteins by isotope tagging (LOPIT)[10,11], SubCellBarCode[12], and Dynamic Organellar Maps (DOMs)[13], which was developed by our lab. The technical differences and individual strengths of each method are extensively reviewed elsewhere[4].

The DOMs approach is particularly powerful for comparative applications, due to its robust fractionation protocol. Briefly, cells are lysed mechanically and the released organelles are partially separated by differential centrifugation[13,14]. Pelleted proteins are quantified across the fractions by MS and the obtained abundance profiles can be

[1]Department of Proteomics and Signal Transduction, Systems Biology of Membrane Trafficking Research Group, Max-Planck Institute of Biochemistry, Martinsried, Germany. [2]School of Biological Sciences, Faculty of Biology, Medicine and Health, Manchester Academic Health Science Centre, University of Manchester, Manchester, UK. [3]Computational Systems Biochemistry Research Group, Max-Planck Institute of Biochemistry, Martinsried, Germany. ✉e-mail: borner@biochem.mpg.de

used to predict protein localization by supervised machine-learning; the result is an organellar 'map of the cell'[13]. Since DOMs are highly reproducible, they allow capture of induced protein localization changes (translocations; Supplementary Note 1) and have thus driven phenotype discovery in diverse biological contexts. For example, DOMs have been applied to reveal the molecular pathomechanisms of AP-4 deficiency syndrome, a severe neurological disorder[15,16]; to characterize the function of a lysosomal retrieval pathway[17]; to quantify translocation events triggered during EGF signalling[13]; to identify the target of drugs selected from a phenotypic screen[18]; and to uncover how HIV infection alters the composition of extracellular vesicles[19]. DOMs can also be easily adapted across different biological sample types, with published studies so far including HeLa cells[13,15–17], a dendritic mouse cell line[18] and primary cortical neurons[20].

While our original DOMs method utilized SILAC (stable isotope labelling by amino acids in cell culture)[21] for protein quantification[13], the application of DOMs beyond cultured cell lines required implementation of different quantification strategies[4,20], including label-free quantification (LFQ; Supplementary Note 1)[22] and the peptide-labelling methods TMT[23] and EASI-tag[24]. SILAC-based maps yield the most precise profiles, but offer limited depth due to increased MS1 spectral complexity (Supplementary Note 1). LFQ maps achieve greater depth but suffer from lower precision, while TMT and EASI-tag maps have intermediate quality[4]. These strategies all applied data-dependent acquisition (DDA; Supplementary Note 1) of MS data[25,26]. The DDA approach only selects high-abundant peptides for identification and quantification, as they are most likely to generate high-quality MS2 spectra (Supplementary Note 1). This simplifies data analysis but introduces a stochastic element to the data collection, leading to inconsistent protein identifications across samples. In the context of DOMs, missing values severely limit the depth of analysis, since profiling requires quantification of the same protein in the majority of measured subcellular fractions and replicates. To alleviate this problem, we previously fractionated peptide samples prior to MS analysis[13,27]. This resulted in improved map depth, but increased MS time requirements.

Owing to recent advances in MS instrumentation and data analysis software, data-independent acquisition (DIA; Supplementary Note 1) is increasingly replacing DDA[28]. In contrast to DDA, the DIA approach does not isolate individual peptides for fragmentation. The resulting MS2 spectra are more complex, which creates technical and computational challenges, but conceptually the approach allows the quantification of all peptides present in a sample. Moreover, unlike in DDA, both the MS1 precursor and the MS2 fragment ions can be used for quantification, which increases precision[29]. DIA is hence becoming the strategy of choice for extensive profiling-based approaches such as SEC-MS[30] and has recently been applied in high-throughput subcellular phosphoproteomics[31].

Here, we harness the power of DIA for generating label-free DOMs. We show that proteomic depth, precision and reproducibility of DIA-DOMs increase dramatically relative to DDA-DOMs, both for generating static maps and in comparative mapping applications. For this purpose, we introduce the software tool DOM-ABC, which enables rapid standardized analysis and quality control of DOMs and other types of profiling data. We provide optimized DIA-DOMs formats with short MS runtimes suitable for high-throughput experiments, and with longer MS runtimes for maximum coverage. Finally, we investigate subcellular rearrangements upon starvation and inhibition of lysosomal acidification in HeLa cells, to demonstrate the power of DIA-DOMs for phenotype discovery.

## Results
### DOM-ABC, a web app for in-depth analysis of profiling data
Our goals were to establish and optimize a DIA-DOMs workflow, and to empower a broad usership of non-proteomic specialists to apply

organellar profiling in biological studies. Therefore, we first implemented DOM-ABC, a software tool for Analysis, Benchmarking and quality Control of dynamic organellar maps. DOM-ABC provides in-depth automated data analysis and interactive info-plots via a graphical user interface (https://domabc.bornerlab.org). As input data, the tool handles raw output files from MaxQuant[32] and Spectronaut[33], as well as any other tabular profiling data. Multiple maps can be uploaded together and directly compared. Firstly, extensive quality control is performed: Proteomic depth is assessed before and after filtering for usable profiles, per map, and across replicates. Principal component analysis (PCA; Supplementary Note 1) plots provide a visual overview of map topology. Two novel metrics are calculated to gauge map quality: 1. Profile scatter of proteins that are part of the same complex ('intra-complex-scatter'; Supplementary Note 1). This reflects within-map profiling precision, based on the assumption that tightly bound proteins co-fractionate with near-identical profiles. 2. Profile scatter of individual proteins across map replicates, which reflects inter-map reproducibility (Supplementary Note 1). Furthermore, subcellular localization is predicted for all profiled proteins by machine-learning, using support-vector machines (SVMs, Supplementary Note 1) based on pre-defined (or customized) compartment marker protein lists provided through the tool. Localization prediction performance is assessed by scoring recall and precision of marker proteins (Supplementary Note 1). Importantly, all of these quality metrics can be benchmarked across experiments and easily compared with reference data to gauge performance over time, across sites or between methods. Lastly, DOM-ABC can perform a differential analysis between multiple maps to identify proteins with altered subcellular localizations (Movement-Reproducibility 'MR' analysis of translocation, Supplementary Note 1[13]). In summary, DOM-ABC provides a user-friendly and automated end-to-end analysis pipeline for spatial proteomics data.

### DIA maps outperform DDA maps across all metrics
First, we extensively optimized DIA-methods for generating label-free DIA-DOMs using a benchmark sample set prepared from three independent subcellular fractionations from HeLa cells, each with six fractions, label-free, as described[20]. We tested a variety of liquid chromatography-mass spectrometry (LC-MS) setups and data acquisition strategies for MS (see Fig. 1 for an overview of the experimental design, and Supplementary Fig. 1 for DIA method optimization). Data were processed with MaxQuant 2.1.3, which features the MaxDIA algorithm[34].

To compare DIA-DOMs to our previous label-free DDA-DOMs workflow, we measured our benchmark samples with DIA and DDA, using a 100 min gradient on a nanoLC. DIA data were processed either with a custom spectral library (ca. 157,000 unique peptide sequences, acquired in DDA mode), or with an in silico spectral library predicted by DeepMassPrism[35] (MaxQuant's 'discovery DIA' mode[34]). PCA plots of the DDA, library DIA and discovery DIA maps looked topologically similar (Fig. 2a). However, DIA maps showed reduced blending of compartment clusters (e.g., mitochondria and Golgi), due to the higher sensitivity of DIA. This leads to better quantification of proteins in fractions where they are of low abundance, resulting in more nuanced profiles (see also Supplementary Fig. 2A).

As expected, the unfiltered proteome depth (Supplementary Note 1) was only slightly increased by DIA (DDA: 6706 protein groups (PGs; Supplementary Note 1); discovery DIA: 7883 PGs; library DIA: 8022 PGs). In contrast, the number of proteins profiled across all three map replicates dramatically increased from 2764 PGs with DDA, to 5866 PGs with discovery DIA (+112%), and to 6571 PGs with library DIA (+138%; Fig. 2b). This performance leap is explained by the much more consistent identification of proteins across samples, reflected by a rise in data completeness from 69% with DDA to 93% with discovery DIA, and to 96% with library DIA (Supplementary Fig. 2B). The proteins

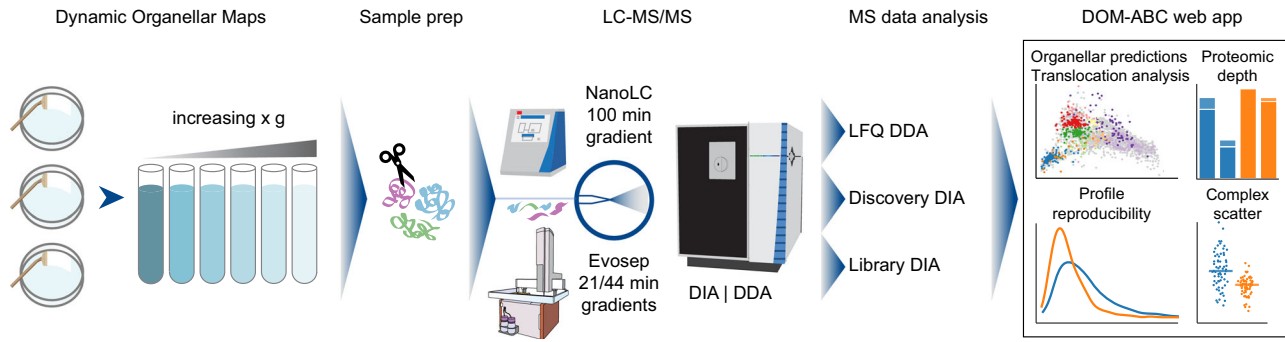

**Fig. 1 | Overview of the Dynamic Organellar Maps workflow, DIA optimization strategy, and evaluation with DOM-ABC.** HeLa cell lysates were fractionated by differential centrifugation, to generate triplicate reference organellar maps. Following tryptic digest, samples were analyzed by LC-MS, using either a Thermo EASY-nLC 1200 HPLC, or a high-throughput Evosep One HPLC, coupled to a Thermo Exploris 480 orbitrap mass spectrometer. Data were acquired in DDA or

DIA mode, and DIA processing was performed with library or discovery DIA. For the performance evaluation, data were analyzed with the web app DOM-ABC (https://domabc.bornerlab.org), which provides multiple info graphics and metrics for assessing map topology, proteomic depth, map resolution and reproducibility, as well as differential analysis for detection of translocating proteins and machine learning based localization prediction.

profiled in the DIA datasets mostly overlapped and contained almost all proteins quantified with DDA (Fig. 2b).

Next, we assessed support-vector machine (SVM)-based compartment classification (Supplementary Note 1), analyzing a set of 839 established marker proteins[13] common to all three maps. DIA maps moderately but clearly outperformed DDA maps (Fig. 2c). Overall F1 score of the hold-out test set (Supplementary Note 1) increased from 91.6% (DDA) to 95.1/93.5% (discovery/library DIA). The combination of increased depth and better SVM performance resulted in a greatly increased number of high and very high-confidence localization predictions, up from 1354 (non-marker proteins) with DDA, to 2884 with discovery DIA (+113%), and to 3204 with library DIA (+136%) (Fig. 2d). In addition, DIA maps provided many more medium-confidence predictions (Fig. 2d). For all three datasets, the concordance with our previously published predictions based on SILAC DOMs[13] was >94% in the high-confidence category (Supplementary Fig. 2C). Finally, we evaluated profile quantification precision and reproducibility (Supplementary Note 1), which are key for comparative spatial proteomics[4]. Within-map profiling precision was markedly improved with DIA (complex scatter reduced by 21% with discovery DIA and by 17% with library DIA; Fig. 2e), and inter-map profile reproducibility was greatly enhanced (profile scatter reduced by 43% with discovery DIA and by 44% with library DIA; Fig. 2f; sample correlations shown in Supplementary Fig. 2D).

Taken together, these data demonstrate that DIA-DOMs strongly outperform our previously established label-free DDA-DOMs with regards to proteomic depth, organellar resolution, precision and reproducibility. Discovery DIA and library DIA provide maps of similar quality, but the measured peptide library boosts depth even further.

**High-throughput LC enables faster and deeper DIA-DOMs**

Based on the excellent depth of DIA-DOMs with 100 min LC gradients, we next evaluated the performance of shorter LC formats, with the aim to establish a fast organellar mapping workflow. We used the Evosep One LC system[36], which runs pre-mixed gradients with standard lengths, e.g., 21 or 44 min, and reduces overhead time between samples to a few minutes. To avoid any confounding effects caused by peptide library generation, we first gauged the performance with discovery DIA. We compared DIA-DOMs run with 100 min (nanoLC), 44 min or 21 min gradients (Evosep), which revealed an almost linear relationship between the number of profiled proteins and runtime (Fig. 3a and Supplementary Fig. 3A, B). Remarkably, the 44 min gradient DIA-DOMs had an increased depth compared to our previous 100 min gradient DDA-DOMs (3,076 PGs profiled across all three map replicates vs. 2,764 PGs with DDA; Fig. 2b), and even with 21 min

gradients, more than 1900 PGs were profiled across three replicates. The compartment prediction performance of the shortest gradient maps was lower (Fig. 3b), but still fairly high in absolute terms (overall F1 = 0.87). The difference was mostly caused by substantial drops in the classifications of three organelles, Golgi, endosome and peroxisomes, which are particularly challenging to resolve in HeLa cells (Fig. 2a, Supplementary Fig. 3C). As expected, shortening the LC gradients also reduced profiling precision (Fig. 3c) and reproducibility (Fig. 3d). Nevertheless, even the shortest (21 min) gradient provided astonishingly well-resolved maps in only around 2.5 h of MS machine time, equivalent to a throughput of over 9 maps per day compared to <2 maps per day using the 100 min nanoLC gradient. Furthermore, processing with a measured peptide library increased the depth of 21 min gradient maps to 3048 PGs (+60%), and the depth of 44 min gradient maps to 4338 PGs (+41%; Supplementary Fig. 3D), with substantial gains in reproducibility (Supplementary Fig. 3E). This makes fast DIA-DOMs even more useful for high-throughput screens and rapid pilot experiments.

We next explored if off-line peptide fractionation and analysis spread over several short LC runs would improve performance relative to a 'single-shot' LC run of equivalent length. Since the Evosep LC system minimizes sample loading overheads, this approach also optimizes the machine-time to gradient-time ratio. We triple-fractionated our benchmark samples by peptide STAGE-tipping[27] and analyzed them with 3 × 44 min LC gradients (with 5 min overheads), which requires little more overall machine time than a single run with a 100 min nanoLC gradient (with 35 min overheads). Remarkably, fractionation yielded ~600 additional profiled PGs relative to the single-shot 100 min gradient (6504 vs 5866 PGs, +11%; Fig. 3a), with similar SVM performance, precision and profile reproducibility (Fig. 3b–d).

We also analyzed maps with 3 × 21 min gradients. Relative to the 100 min gradient, this reduced machine time by ~50%, yet incurred only a moderate drop in performance (Fig. 3a–d). Thus, the 3 × 21 min format provides a useful compromise between speed and quality. Finally, we ran fractionated samples with 3 × 100 min nanoLC gradients, to provide extra-deep coverage (7443 PGs, +48% vs. single-shot 100 min discovery DIA). Of note, these are the deepest organellar maps from HeLa cells to date, providing a rich resource for protein subcellular localization predictions (Supplementary Data 1).

In conclusion, our data show that deep, high-accuracy DIA-DOMs can be prepared with short LC gradients, enabling high-throughput spatial proteomics. In conjunction with off-line peptide fractionation, single long gradients can be replaced with multiple short gradients, optimizing machine time and enhancing map depth even further.

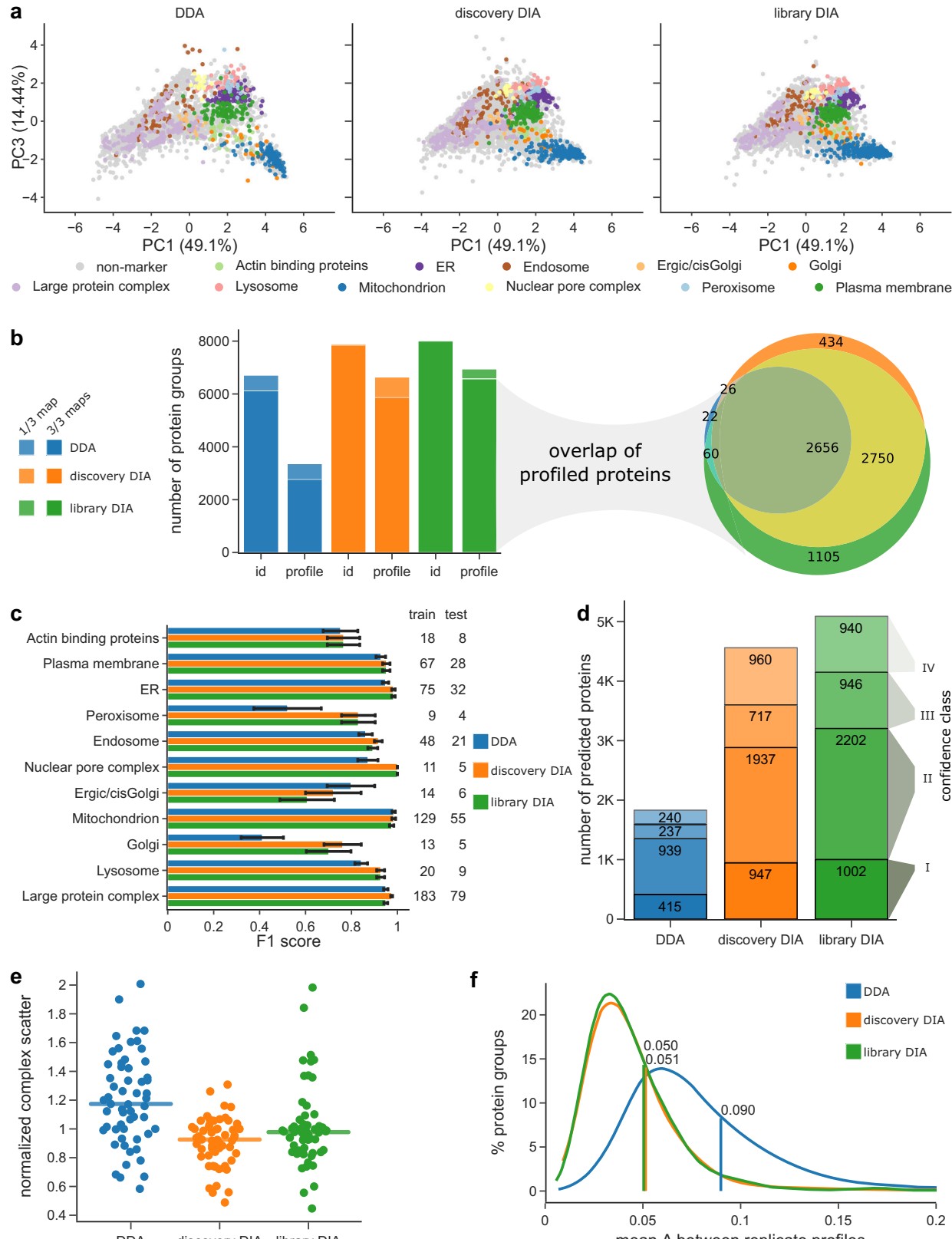

### DIA-DOMs reveal effects of starvation/BafA treatment in HeLa

We next tested the capabilities of DIA-DOMs for detecting induced subcellular localization changes. Nutrient deprivation in combination with Bafilomycin A1 (BafA) treatment is a widely used method for investigating autophagy[37]. While the starvation induces metabolic changes including autophagy[38], the BafA treatment increases endo-lysosomal pH by inhibiting vATPase function and thus prevents lyso-somal protein degradation[39,40]. This helps to gauge autophagic flux and facilitates the capture of autophagic structures by imaging[37]. However, the endosome is a major protein trafficking hub, and increasing lumenal pH blocks endosomal exit pathways. As a result, proteins that normally cycle between endosomes and the plasma

**Fig. 2 | Comparison of DDA, discovery DIA and library DIA-based maps.** All maps were acquired with 100 min LC gradients. Source data are provided as a Source Data file. **a** Topology of organellar maps in PCA space. Coloured dots correspond to organellar marker proteins. For comparability a single PCA was performed across all three experiments and the comparative experiments displayed in Fig. 4. PCs 1 and 3 provide the best visual separation of non-nuclear compartments, as PC2 (30% variability) is dominated by nuclear proteins. **b** Left panel: Number of proteins identified or profiled in at least 1 or in all 3 out of 3 replicate maps. Right panel: Overlap of profiled proteins between acquisition modes. **c** Performance of support vector machine compartment classification. The same 839 marker proteins were used for all three maps. The numbers of markers used for training (70%) and testing (30%) are indicated for each compartment. F1 scores are the harmonic mean of recall (true positives / [true positives + false negatives]) and precision (true positives / [true positives + false positives]) based on the test set. Error bars show the standard deviation of 20 sub-samples from the test set. **d** Number of organellar assignments, by confidence class - (I) very high, (II) high, (III) medium, (IV) low (see Methods). The 839 marker proteins are not included. **e** Normalized profile scatter within stable protein complexes. Only non-redundant complexes with at least five subunits quantified across all datasets were included. Each point represents the average normalized distance to the median complex profile in one map replicate. Horizontal lines indicate the median from $n = 57$ quantifications across 3 biological replicates. **f** Inter-replicate scatter, for the 2656 proteins profiled across all conditions and replicates. X-axis shows the average absolute distance of replicates to the corresponding average protein profile. The lines and numbers indicate the 70th percentile. Lower scatter reflects higher map reproducibility (X-axis cut at 0.2; <1% of profiles not shown).

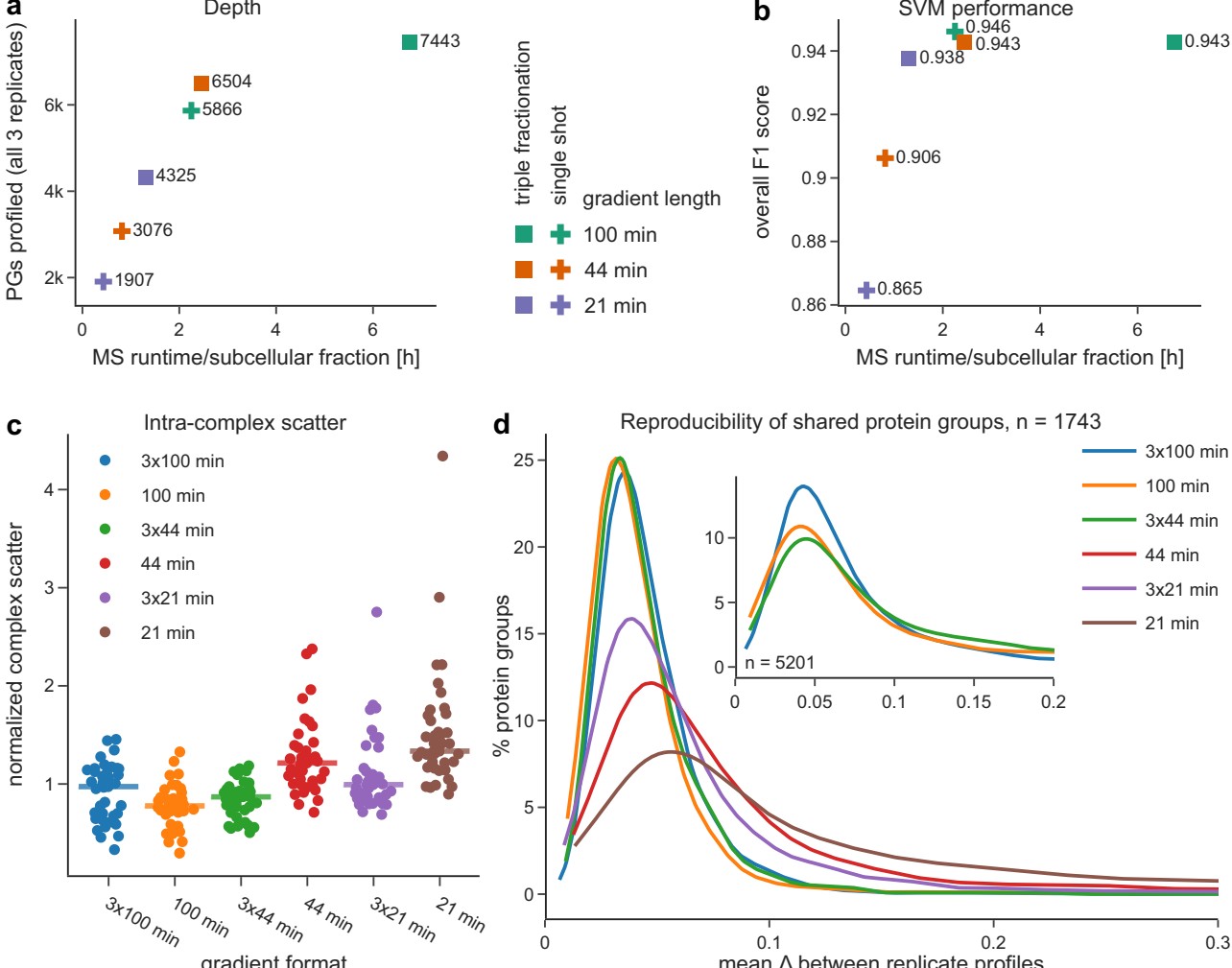

**Fig. 3 | Comparison of DIA-DOMs performance with different LC gradients and sample fractionation.** Comparison of maps measured with a 100 min nanoLC gradient, or with 44 min / 21 min gradients on the Evosep One LC system. SDB-RPS STAGE-tipping was employed for triple-fractionation. All maps were analyzed with discovery DIA. Source data are provided as a Source Data file. **a** Proteomic depth after filtering for profile completeness across three replicates in relation to the MS runtime investment. **b** SVM classification performance (overall marker F1 score) in relation to the MS runtime investment. For the single-shot 100 min and the triply-fractionated 44 min and 100 min maps, the same 988 organellar marker proteins were used. For shorter gradient single-shot maps and the triply-fractionated 21 min maps, only the 637 markers overlapping between these were used. **c** Normalized intra-complex scatter, quantified by the average absolute distance to the median complex profile. Only complexes with at least five subunits quantified across all datasets were included. Points represent scatter in individual replicate measurements relative to the median across all experiments. The median of 39 quantifications across 3 biological replicates is indicated. **d** Inter-replicate scatter for the 1743 proteins quantified across all datasets. This number was limited by the depth of the 21 min dataset. (X-axis cut at 0.3; <8% of profiles not shown.) Inset: equivalent plot for the 5201 proteins quantified across the 100 min, 3 × 44 min, and 3 × 100 min datasets.

membrane, or between endosomes and the Golgi, become trapped in endosomes[41,42]. This 'side effect' of BafA treatment is largely ignored in investigations of autophagy, but may have considerable bearings on the interpretation of results. Moreover, it is not generally known which proteins get trapped, as only relatively few have been identified to date[17,41–43]. Here, we applied DIA-DOMs for a global analysis of subcellular localization changes induced by starvation and BafA treatment.

We prepared triplicate DIA-DOMs (100 min gradient, library DIA) and full proteomes from HeLa cells that were either starved for 1 h in the presence of BafA, or left untreated. Evaluation with DOM-ABC showed that the two conditions yielded topologically very similar maps (Fig. 4a). The strength of DIA-DOMs was highlighted again by the remarkable profiling depth (>5800 PGs in each condition) and the almost complete overlap of profiled proteins (Supplementary Fig. 4A).

To identify proteins with altered subcellular localizations, we performed our previously established movement and reproducibility (MR) analysis[13,44], now implemented in DOM-ABC with additional reproducibility filters and automated downstream processing (see Methods). Since starvation/BafA treatment drastically reprograms cellular metabolism, we expected pleiotropic subcellular rearrangements, in addition to endosomal trapping. We identified 164 proteins with significant localization shifts (False discovery rate (FDR) < 5% (Supplementary Note 1)) (Fig. 4b; Supplementary Fig. 4B; Supplementary Data 2). Of these, 142 were also quantified in our full proteomes (Supplementary Fig. 4D), and none changed significantly in abundance. Thus, our MR analysis specifically revealed proteins that respond to starvation and BafA treatment by subcellular relocalization. To categorize hits functionally, we performed hierarchical clustering (Supplementary Note 1) on the profile changes and identified seven main groups (Fig. 4c). Five of these groups were highly enriched in specific functions or localizations (Supplementary Fig. 4C). One cluster (red) contained 10 Golgi proteins (shown in black in Fig. 4b), as well as several other transmembrane and soluble secretory pathway proteins. A second cluster (purple) was enriched in endosomal and late secretory pathway proteins (Fig. 4c). Inspection of Golgi protein profiles in untreated vs. treated cells revealed that these shifted towards an endosomal profile, consistent with endosomal trapping (Fig. 4d, and interactive Supplementary Data 3).

The largest cluster (green) predominantly contained proteins involved in translation and mRNA processing, including over 50 core ribosomal proteins, which shifted towards the highest speed (80 K) fraction (Fig. 4c)). This is consistent with a starvation-induced suppression of protein translation, resulting in an increased number of free ribosomes or smaller translational assemblies[45]. Strongly supporting this hypothesis, we observed an almost two-fold increase of these proteins in the cytosolic fraction (Supplemental Fig. 4E; $p = 1.4E$ −30). A fourth cluster (blue) was enriched in transcriptional regulators. Two of them, FOXK1 and FOXK2, were reduced in the nuclear fraction (Supplementary Fig. 4F) and increased in the cytosol (Supplementary Fig. 4E), consistent with their known function as negative transcriptional regulators of autophagy[46]. The fifth cluster (orange) included several mitochondrial proteins with a non-mitochondrial pool that appears to be lost after starvation/BafA treatment (Supplementary Data 3); mitochondria are known to be extensively modified during starvation[47]. The hits in Fig. 4b, c represent only the most prominent translocation events in our dataset; therefore, we created an interactive supplementary table for the detailed exploration of individual profile shifts (Supplementary Data 3).

Next, we re-acquired treated and control maps with DDA and repeated the analysis (Supplementary Fig. 5). As expected, DIA-DOMs provided much greater reproducible profiling depth (4475 vs 2055, Supplementary Fig. 5A), identified a greater number of significant translocations (164 vs 128, Supplementary Fig. 5B-D), and obtained better scores in the MR analysis (Supplementary Fig. 5E) as well as

quality metrics (Supplementary Fig. 5F) than DDA-DOMs. This demonstrates that DIA-DOMs also provide superior performance for comparative applications and reveal biological insights that would be missed with DDA-DOMs (Supplementary Fig. 5G-J).

Taken together, our DIA-DOMs analysis revealed a broad spectrum of established and previously unknown subcellular rearrangements related to gene regulation and metabolism induced by starvation and BafA treatment, as well as endosomal trapping of diverse endomembrane proteins. Intriguingly, we identified ten Golgi proteins that shifted completely or partially towards endosomes (Fig. 4b). This group contained some of the strongest hits in the MR analysis (e.g., GLG1 and TM9SF2, Fig. 4b, d), prompting us to characterize the behaviour of Golgi proteins in more detail.

## DIA-DOMs detect cycling Golgi proteins with diverse kinetics

Endosomal trapping in response to increased endo-lysosomal pH has been reported previously for a small number of Golgi proteins[17,41–43], but it has not been studied systematically. Such an analysis could distinguish Golgi proteins which undergo anterograde cycling from those that are relatively static, which would shed light on a fundamental feature of Golgi homeostasis (Fig. 5a). The ten endosome-trapped Golgi proteins identified above (Fig. 4b) differed considerably in shift magnitudes (Fig. 4b, d), indicating differential degrees of trapping. This also suggested that there may be further Golgi proteins with partial shifts below the detection limit of the MR analysis. For a comprehensive and systematic characterization of Golgi protein behaviour, we developed a targeted correlation analysis strategy. We first compiled a list of 42 transmembrane and 2 lumenal Golgi proteins from our untreated maps. We then calculated all pairwise correlations of their profile shifts and performed hierarchical clustering. Strikingly, three clearly segregated groups emerged from the data (Fig. 5b). The first group (Cluster 1; 22 proteins) contained all ten proteins identified by MR analysis, which formed a particularly well-defined core. This group also included the two lumenal Golgi proteins (SDF4 and FAM3C), suggesting they too may be subject to endosomal trapping. To characterize cluster behaviour in detail, we plotted the translocations of all proteins in PCA space. Cluster 1 proteins showed largely parallel translocations of different magnitudes from Golgi to endosomes/lysosomes (Fig. 5c). In contrast, Cluster 2 proteins showed very small shifts within the Golgi boundaries (Fig. 5d). The third group had more variable shifts, but these did not indicate translocation towards endosomes, and remained confined to the Golgi (Supplementary Fig. 6A). Thus, our data reveal two classes of Golgi proteins−those that undergo partial or complete endosomal trapping (Cluster 1, cycling) and those that do not show any trapping under the experimental conditions (Clusters 2 + 3, static). To assess the individual degree of trapping for proteins in Cluster 1, we ranked them by magnitude of translocation towards endosomes (Fig. 5e). The top proteins mostly corresponded to the hits from our MR analysis. Importantly, the remaining proteins included GOLM1, which has previously been shown to undergo endosomal trapping, but was not detected by the MR analysis[43]. Evaluating these endosomal shifts relative to the distribution of all endosomal shifts in the dataset (Supplemental Fig. 6B) allowed us to classify translocations as complete, partial or negligible. We thus identified 15 Golgi proteins that we predict undergo endosomal trapping to varying degrees (Fig. 5e). As above, we then repeated the analysis with the equivalent set of DDA-DOMs. The DDA dataset also yielded the split into cycling and static proteins, but covered only 20 Golgi proteins, of which only 8 were identified as cycling proteins (Supplementary Fig. 7). Thus, as expected, DIA-DOMs provided a much more comprehensive analysis.

Next, we used immunofluorescence microscopy to validate our predictions. Prior to fixation, cells were cultured for 1 h in either full or starvation medium, with or without BafA. Labelling for the

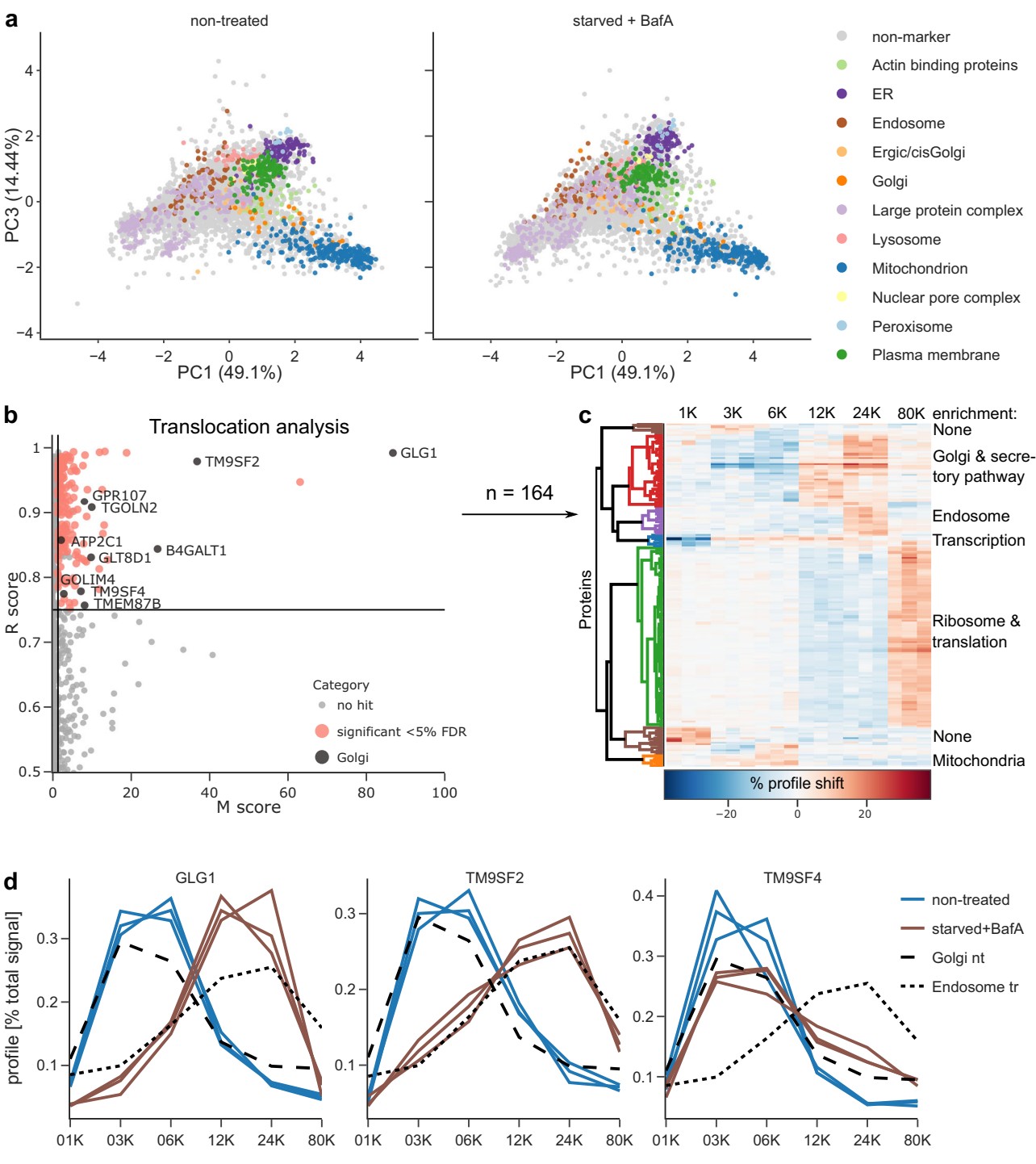

**Fig. 4 | DIA-DOMs reveal the cellular effects of starvation and Bafilomycin A1 treatment.** HeLa cells were either left untreated, or starved in the presence of 100 nM BafA for 1 h. DIA-DOMs were prepared in triplicate. **a** PCA maps show overall similar topology in both conditions. Source data are provided as a Source Data file. **b** Movement-Reproducibility (M-R) analysis detects 164 proteins with significantly altered subcellular localization (marked in red and black) at an FDR < 5%, which corresponds to an M-score cut-off at 1.3. The M-score is the −log10 of the Benjamini-Hochberg corrected combination of *p*-values derived from three independent chi2 distributions of robust Mahalanobis distances (see Methods for details). In addition, the R-score cut-off was 0.75, and proteins with only one replicate *p*-value < 0.1 were excluded. Integral Golgi membrane proteins with significant M-R scores are marked in black. **c** Hierarchical clustering of the 164 significant profile shifts by Pearson correlation. Cluster annotation is based on GO-term annotation enrichment (see Supplementary Fig. 4C); the brown clusters show no significant enrichments at 10% FDR. As expected, we did not observe the re-localization of core autophagic machinery, since HeLa cells already have high basal levels of autophagy (see Supplementary Fig. 6C). **d** Profiles of three significantly shifting Golgi proteins, overlaid with average profiles for Golgi markers (non-treated, nt) and endosome markers (starved/BafA treated, tr). GLG1 and TM9SF2 show complete transitions from Golgi to endosomes, and TM9SF4 only a partial transition.

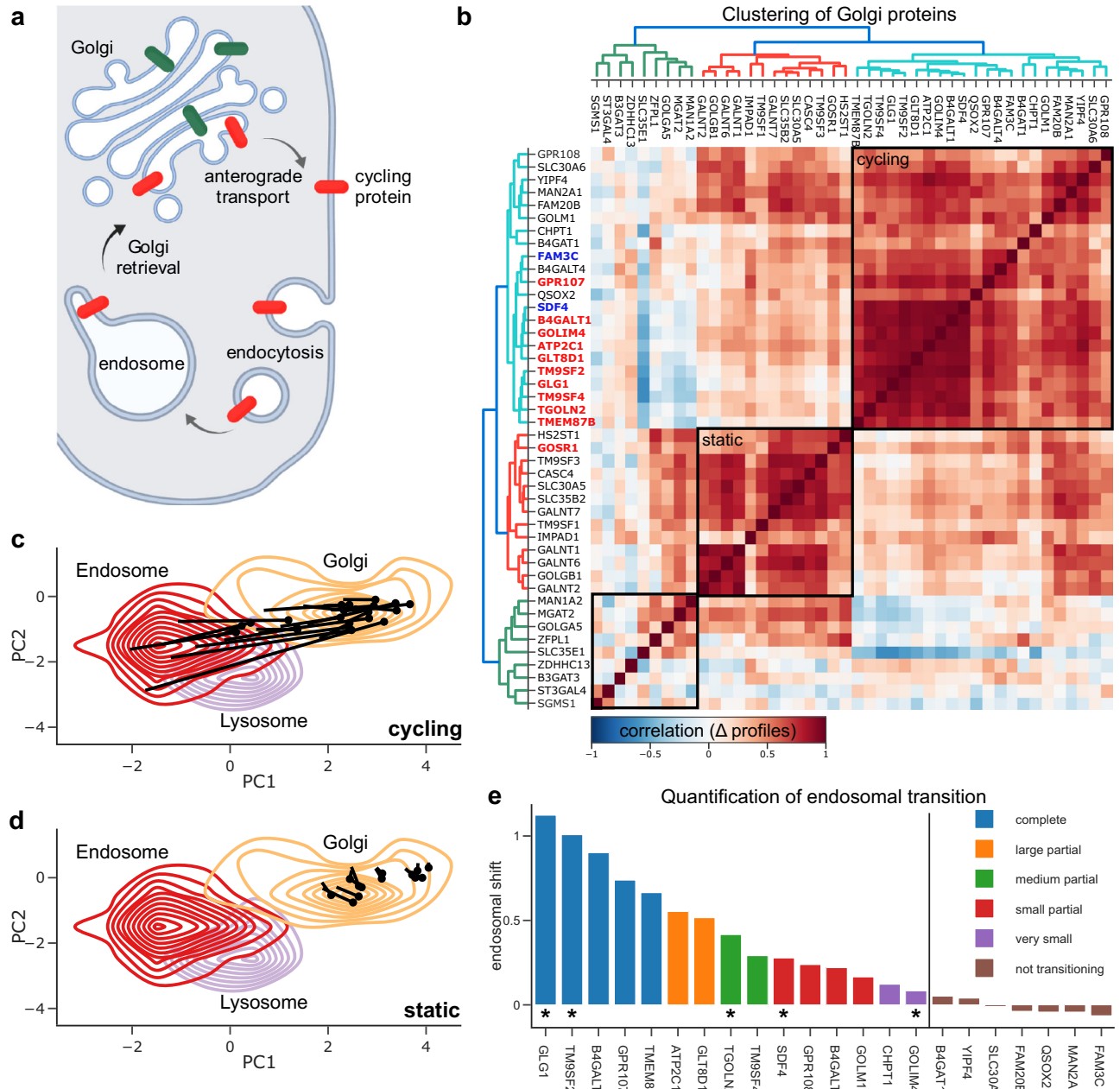

**Fig. 5 | In-depth analysis of Golgi protein behaviour caused by starvation and Bafilomycin A1 treatment.** Source data are provided as a Source Data file. **a** Some Golgi proteins cycle to the plasma membrane and back, via endosomes (red). BafA treatment compromises the retrieval pathway, trapping Golgi proteins in endosomes; non-cycling Golgi proteins (green) remain unaffected. Created with BioRender.com. **b** Correlation matrix of delta profiles and clustering of integral membrane and lumenal Golgi proteins. Gene name colours: red = primary hits from MR outlier analysis, blue = lumenal proteins. Boxes highlight the three sets of proteins shown in panels (**c**) (cycling), (**d**) (static) and Fig. S6A (also static). **c, d** Protein trajectories starting at untreated positions (pin head) overlaid with contour density plots of relevant organellar marker proteins. These are calculated from the PCA coordinates of the non-treated condition also shown in Fig. 4a. Please

note that axes are scaled by PC variance, for faithful visualization of translocations in PCA space. **c** Anterograde cycling Golgi proteins (Cluster 1) shift towards the endosome/lysosome. **d** Static Golgi proteins (Cluster 2) stay within the core density of the Golgi apparatus. **e** The relative shift towards an average endosomal profile stratifies phenotype strength. The plot shows Cluster 1 proteins identified in (**b**), in order of observed shift magnitude. The vertical line indicates the cut-off below which the shift was considered negligible. Proteins indicated with an asterisk were selected for further validation by imaging (Figs. 6, 7). Categorization into complete, partial and small shifts was based on the increase in correlation with the average endosomal profile, relative to endosomal shifts observed in the whole proteome (Supplementary Fig. 6B).

autophagosome marker LC3B confirmed that the treatments worked as expected (Supplementary Fig. 6C). In response to BafA treatment, GLG1 showed a complete transition from the Golgi to a punctate endosomal pattern (Fig. 6a) and TGOLN2 (also known as TGN46) underwent a partial translocation out of the Golgi (Fig. 6b), as predicted by our DIA-DOMs analysis. The translocations occurred in

response to BafA, in both full and starvation medium, but not in starved cells in the absence of BafA. This confirmed our hypothesis that endosomal trapping in our DIA-DOMs experiment was caused by BafA treatment, and not by starvation. As expected, GALNT2 retained its Golgi pattern under all conditions (Fig. 6a, b), suggesting it is a static Golgi protein.

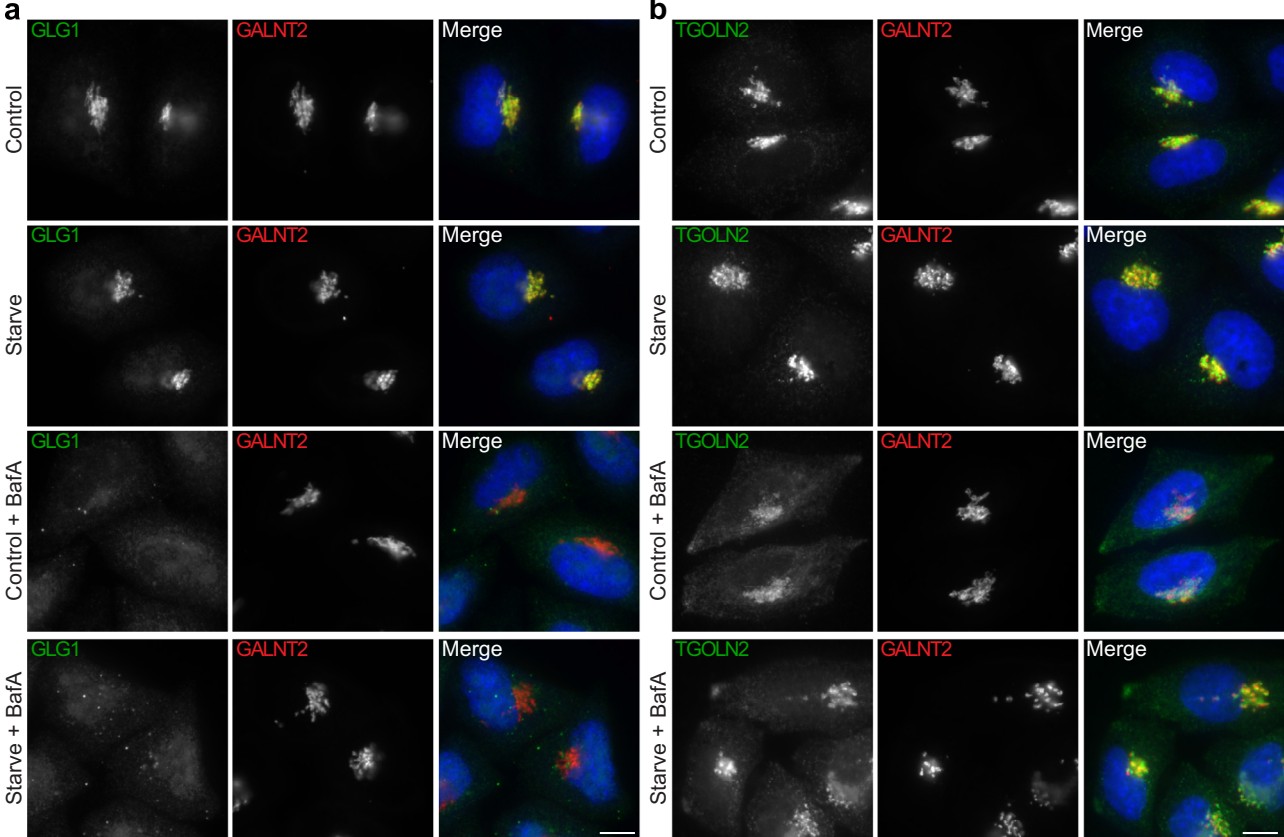

**Fig. 6 | Bafilomycin A1 (BafA) treatment causes a relocalization of GLG1 and TGOLN2 away from the Golgi.** Widefield imaging of immunofluorescence labelling of GLG1, TGOLN2 (TGN46), and GALNT2 validated the localization shift predictions shown in Fig. 5. HeLa cells were cultured for 1 h in either: (1) full growth medium (Control); (2) EBSS to starve the cells (Starve); (3) full growth medium plus 100 nM BafA (Control + BafA); or (4) EBSS plus 100 nM BafA (Starve + BafA). **a** GLG1 (green) completely disperses from the Golgi in BafA treated cells, with or without starvation; **b** TGOLN2 (green) shifts away from the Golgi, with or without starvation; **a, b** GALNT2 (red) remains unchanged. In the merged images, DAPI labelling of the nucleus is also shown (blue). Scale bars: 10 μm. Note, the relocalization effects are caused by BafA treatment and not by starvation. Images are representative of at least 13 images per condition, from two separate imaging experiments. Brightness levels for display were adjusted individually for each condition, to avoid saturation and aid visualization of signal distribution. Raw data have been uploaded to https://zenodo.org/record/8197844.

Differences in the magnitude of translocation of individual Golgi proteins might reflect different cycling kinetics, or different proportions of the mobile pool of a protein. To investigate this further, we selected six proteins with different profiling behaviours for a detailed imaging-based analysis: GLG1 and TM9SF2 (for which we predicted complete translocations to endosomes after 1 h); TGOLN2, SDF4, and GOLIM4 (for which we predicted partial endosomal translocations after 1 h; Fig. 5e), and GALNT2 from the predicted static Cluster 2. We treated HeLa cells with BafA and followed the journey of these six proteins by immunofluorescence microscopy over a time-course of eight hours (Fig. 7). First, we evaluated qualitative changes of the staining patterns. As predicted by our DIA-DOMs after 1 h BafA treatment, the GALNT2 Golgi pattern remained completely unchanged throughout the time-course, even after 8 h (Fig. 7a). Thus, GALNT2 can be considered a static resident of the Golgi. In contrast, GLG1, TM9SF2, TGOLN2, SDF4 and GOLIM4 all changed localization from a predominantly Golgi pattern to a more peripheral punctate pattern, consistent with endosomal trapping, but with differing dynamics over the time-course (Fig. 7c, e, g, i, k). To quantify the translocations at each timepoint, we next developed an automated image analysis pipeline to quantify the fluorescence signal in the Golgi (marked by static GALNT2) relative to the rest of the cell (marked by phalloidin) (Fig. 7b, d, f, h, j, l). This revealed diverse movement behaviours for the different Golgi protein targets. GLG1 and TM9SF2 underwent rapid and complete transitions to endosomes within 1 h, with most of the protein

having exited the Golgi after just 30 min (Fig. 7c–f). While TGOLN2 showed equally fast endosomal transition in around 30 min to 1 h, this plateaued rapidly, with a significant pool of TGOLN2 remaining in the Golgi until the last timepoint (Fig. 7g, h). This suggests that TGOLN2 maintains a relatively immobile pool at the Golgi. GOLIM4 showed steady but relatively slow movement away from the Golgi, which was detectable after 1 h but still ongoing after 8 h (Fig. 7i, j). The change in distribution of the lumenal protein SDF4 was even more subtle and only became detectable after 2 h, with a gradual decrease in Golgi signal proceeding until the end of the time-course (Fig. 7k, l).

These data reveal that cycling Golgi proteins follow remarkably different patterns and kinetics. Importantly, our DIA-DOMs data correctly predicted which proteins would show a complete transition at 1 h, and which proteins would show strong or weak partial transitions (compare Fig. 7 with Fig. 5e). Remarkably, for SDF4 the quantitative imaging data did not yet indicate a significant change at 1 h (only from the next timepoint on); in contrast, our proteomic profiling data already predicted a weak partial transition at 1 h, demonstrating the superior sensitivity of comparative DIA-DOMs for detecting subtle changes in protein subcellular localizations.

In conclusion, our targeted correlation-based analysis increases the sensitivity of Dynamic Organellar Maps for detecting small, but highly correlated shifts, and enables prediction of relative phenotype strength. Our systematic assessment of Golgi protein anterograde cycling behaviours illuminates an important aspect of Golgi organelle

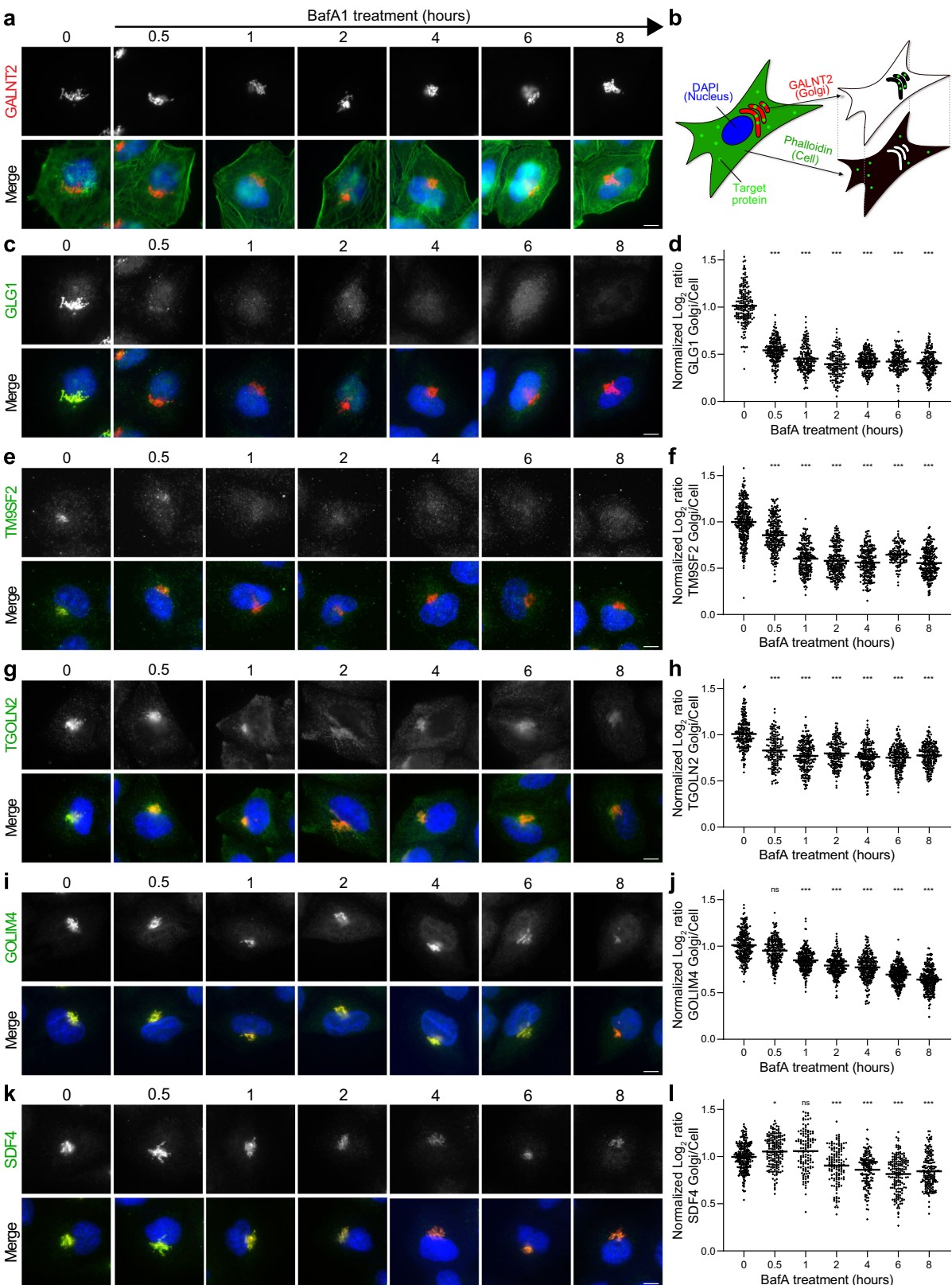

homeostasis, and demonstrates the power of DIA-DOMs for functional investigations.

## Discussion

Dynamic organellar maps (DOMs) capture protein localizations and their changes at the proteome scale, and the approach has driven diverse discoveries in cell and medical biology[13,15–19]. The bottleneck of our original DDA-based workflow was the considerable MS time required to achieve deep coverage, which created a barrier for non-specialist labs to use the approach. Here, we introduce label-free DIA-DOMs, which overcome this limitation. DIA-DOMs achieve twice the proteomic depth of label-free DDA-DOMs in the same MS time and strongly improve map performance across all metrics, in static and comparative applications. Conversely, DIA-DOMs achieve the same

**Fig. 7 | Quantitative microscopy of an 8-hour time-course of Bafilomycin A1 (BafA) treatment demonstrates differing cycling behaviour of Golgi proteins.** HeLa cells were left untreated in full growth medium (0 h) or were cultured in the presence of 100 nM BafA for 0.5, 1, 2, 4, 6 or 8 h, before fixation, immuno-fluorescence labelling and semi-automated widefield imaging. Brightness levels for display of representative images were set uniformly across conditions for each channel. **a** Representative images of cells labelled with anti-GALNT2 (red). In the merged image, phalloidin-488 labelling of actin (green) and DAPI labelling of the nucleus (blue) are also shown. Scale bar: 10 μm. GALNT2 localization at the Golgi does not change over the 8 h time-course. Note, these are the same cells as shown in panel (**c**). **b** Schematic detailing our automated image analysis pipeline for determining the distribution of target proteins between the Golgi (GALNT2 region) and the rest of the cell (phalloidin region). A decrease in the corresponding ratio indicates that a target protein is moving out of the Golgi into other parts of the cell.

**c**, **e**, **g**, **i**, **k** Representative images of cells labelled with **c** anti-GLG1, **e** anti-TM9SF2, **g** anti-TGOLN2, **i** anti-GOLIM4, or **k** anti-SDF4 (all green). In the merged image, anti-GALNT2 labelling of the Golgi (red) and DAPI labelling of the nucleus (blue) are also shown. Scale bars: 10 μm. Note, the target proteins predominantly appear in the Golgi at 0 h, demonstrated by overlap with GALNT2, but their signal in the Golgi decreases as the time-course progresses. **d**, **f**, **h**, **j**, **l** Quantification of the ratio of **d** GLG1, **f** TM9SF2, **h** TGOLN2, **j** GOLIM4, or **l** SDF4 labelling intensity between the Golgi and the rest of the cell, over the 8 h time-course. Source data are provided as a Source Data file. Each datapoint indicates the normalized log2 ratio for an individual cell (bar indicates median; $n \geq 125$ cells per condition examined over two independent experiments). A Kruskal–Wallis test with Dunn's post-test was performed for comparisons to the 0 h timepoint: ***$p \leq 0.001$; **$p \leq 0.01$; *$p \leq 0.05$; ns $p > 0.05$. See statistics and reproducibility section and Supplementary Table 1 for further details.

depth as DDA-DOMs in a small fraction of the MS runtime. Importantly, the instrumentation required for DIA-MS is the same as for DDA-MS, which facilitates easy adoption of the technique.

Based on our extensive optimization, we now recommend the following formats for DIA organellar mapping: (1) Deep mapping with 12 h of MS time per map, either using single runs on a 100 min nanoLC gradient or fractionation/triple runs with 44 min Evosep gradients, recommended for biological exploration; (2) High-throughput maps with ~2.5 h MS time per map, using Evosep 21 min gradients, recommended for pilot experiments or more complex experimental designs. Suitable peptide libraries generated by additional DDA measurements further boost the depth of either format, but are optional, due to the power of in silico predicted DIA libraries/discovery DIA. While our original DDA-SILAC DOMs[13,20] still provide the highest quantification precision (lowest intra-complex-scatter) due to the accuracy of SILAC-based quantification, DIA-DOMs are just as reproducible and offer three times greater proteomic depth in the equivalent MS runtime (Supplementary Fig. 6D–F). For the majority of applications DIA-DOMs will thus outperform even SILAC-based DDA-DOMs. However, where there is a need to detect very small protein translocations in systems that allow metabolic labelling, DDA-SILAC DOMs may still offer an advantage. Future studies should investigate if DIA-SILAC[48] can be applied to DOMs to further enhance the performance.

We created DOM-ABC (https://domabc.bornerlab.org), an open-source web app for the analysis of profiling data. DOM-ABC was crucial for our detailed and objective comparisons of different DOMs workflows. Importantly, with the introduction of DOM-ABC, we now provide a complete and seamless workflow for spatial proteomics from bench to data visualization, which can be carried out by labs without specialist proteomics expertise. DOM-ABC has several unique features, including the ability to perform comparative translocation analyses, using our MR method, at the click of a button. To our knowledge, it is the only tool that provides a comprehensive end-to-end analysis pipeline for spatial proteomics data that is completely accessible through a graphical user interface. Unlike the only other published proteomic profiling analysis package, pRoloc[49], DOM-ABC requires no programming skills or software downloads. Furthermore, DOM-ABC offers interactive visualization of all quality metrics and results, which can be exported as publication-ready figures (Figs. 2, 3 and 4 are mainly based on plots provided by DOM-ABC). Importantly, all metrics and visualizations are generated automatically, and do not require expert user input. Previously published quality assessment tools for spatial proteomics data focus on single aspects, such as compartment prediction performance (MetaMass[50]) or organellar resolution (Qsep[51]). In contrast, DOM-ABC covers a broad range of metrics, including novel metrics for the assessment of profiling precision and reproducibility (Supplementary Note 1), and also allows the integration of MetaMass[50], Perseus[52], and other external classification data. DOM-ABC thus enabled us to perform unbiased DIA method development and objective method comparisons. Importantly, DOM-ABC accepts

custom input data not restricted to a particular software or profiling method, and will thus greatly facilitate streamlined and standardized analysis of spatial proteomics experiments. Since the code is publicly available and provides usage examples and interfaces beyond the graphical user interface, any python-versed user can also expand and customize the generated figures and the tool itself.

By achieving highly reproducible organellar profiles, the original DOMs approach enabled MS-based comparative spatial proteomics for the first time[13]. Today, several global organellar profiling approaches are firmly established, including LOPIT, PCP, SubCellBarCode, and DOMs[3,4,9–12,31,53]. All provide high-quality organellar maps and have unique features and individual advantages, which are reviewed in detail elsewhere[3,4]. Of these methods, our label-free DOMs have the simplest workflow (fractionation by centrifugation) and require the least MS runtime. We would hence argue that DOMs currently offer the easiest option for labs venturing into spatial proteomics. Our extensive protocols[44], in conjunction with our DOM-ABC tool, further facilitate rapid method establishment. Of note, the Olsen/Lund-Johansen labs recently introduced a fast spatial proteomics method based on chemical fractionation[31]. While our fastest DIA-DOMs format shares some operational similarities (6 fractions, 21-min Evosep gradients, DIA), the two methods are conceptually different, as DOMs investigate intact organelles, whereas chemical fractionation necessitates organellar lysis. As a result, chemical fractionation achieves high resolution of sub-nuclear compartments, but at the cost of much lower resolution of membrane-bound organelles; the two methods thus offer partially complementary insights. Nevertheless, a detailed side-by-side comparison shows that our fastest DIA-DOMs format provides superior proteomic depth, precision and compartment prediction performance, when the same standardized DOM-ABC analysis workflow is applied (Supplementary Fig. 8).

To test DIA-DOMs for phenotype discovery, we assessed protein localization changes upon nutrient starvation in the presence of Bafilomycin A1. This treatment is routinely used to investigate autophagy, but also blocks protein exit from endosomes, which causes a poorly characterized and often disregarded traffic jam in the endomembrane system. Our analysis mapped over 160 protein localization changes associated with starvation and metabolic reprogramming, as well as extensive endosomal trapping of secretory pathway proteins, which can be explored through our interactive Supplementary Data 3. Intriguingly, many of the endosomally trapped proteins normally reside in the Golgi. To further dissect how Golgi homeostasis is affected, we performed a targeted profile shift analysis of all Golgi proteins captured by our maps. This revealed two populations: proteins that undergo endosomal trapping and proteins with persistent Golgi localization. This observation is consistent with a model in which some Golgi proteins undergo anterograde cycling via endosomes, and others do not, as previously proposed[41]. Our data now substantially expand this model. First, we provide a systematically derived compendium of 15 cycling Golgi proteins. Furthermore, we observe

pronounced differences in the degree of endosomal trapping within the experimental timeframe. Phenotypic strength varied from complete transitions (e.g., GLG1), to partial (e.g., TGOLN2/TGN46) and very subtle shifts (e.g., GOLIM4). To further characterize cycling phenotypes, we performed a quantitative imaging analysis over an 8 h time-course of BafA treatment. This revealed highly individual cycling patterns, with different kinetics for Golgi depletion ranging from the rapid loss of GLG1 and TM9SF2, to the much slower departure of SDF4 and GOLIM4. In the case of TGOLN2, our image analysis also provides evidence for an apparently immobile Golgi pool, in addition to the cycling pool. Remarkably, our DOMs analysis correctly predicted phenotypic strength and, in the case of SDF4, even surpassed the sensitivity of quantitative microscopy. While our data support the existence of a subset of static Golgi proteins, it is possible that these may also cycle, either with very slow kinetics not detectable in the 8 h time-course, or under different physiological conditions; alternatively, the cycling route may bypass the endosome and thus be insensitive to BafA treatment. While we currently cannot distinguish between these scenarios, our identification of large sets of apparently static and cycling proteins, and the prediction of their relative cycling speeds, will facilitate future investigations into this fundamental property of Golgi proteins.

In conclusion, DIA-DOMs enable label-free organellar profiling with high depth, speed and precision, and provide a powerful tool for systematic phenotype discovery.

## Methods

### Experimental protocols

**Antibodies.** The following antibodies were used in this study: mouse (IgG2b) anti-GALNT2 1:200 for IF (BioLegend Cat# 682302, RRID:AB_2566611), rabbit anti-GLG1 1:200 for IF (Sigma Aldrich Cat# SAB1303679), mouse (IgG1) anti-GOLIM4 1:1000 for IF (Enzo Life Sciences Cat# ALX-804-603-C100, RRID:AB_2051552), mouse anti-LC3B 1:400 for IF (MBL International Cat# M152-3, RRID:AB_1279144), rabbit anti-SDF4 1:400 for IF (Sigma Aldrich Cat# HPA011249, RRID:AB_2668468), sheep anti-TGN46 (TGOLN2) 1:200 for IF (Bio-Rad Cat# AHP500, RRID:AB_324049) and rabbit anti-TM9SF2 1:200 for IF (Abcam Cat# ab271123). Fluorescently labelled secondary antibodies were purchased from Thermo Fisher Scientific and used at 1:500 for IF: Alexa Fluor 488-labelled donkey anti-mouse IgG (Cat# A-21202, RRID: AB_141607), Alexa Fluor 555-labelled goat anti-rabbit IgG (Cat# A32732, RRID:AB_2633281), Alexa Fluor 568-labelled donkey anti-rabbit IgG (Cat# A10042, RRID:AB_2534017), Alexa Fluor 555-labelled donkey anti-sheep IgG (Cat# A-21436, RRID:AB_2535857) and Alexa Fluor 680-labelled donkey anti-sheep IgG (Cat# A-21102, RRID:AB_2535755). For co-labelling of GALNT2 and GOLIM4, isotype-specific anti-mouse secondaries were used: Alexa Fluor 555-labelled Goat anti-Mouse IgG1 1:500 for IF (Cat# A-21127, RRID:AB_2535769) and Alexa Fluor 647-labelled Alpaca anti-Mouse IgG2b Nano (VHH) Recombinant Secondary Antibody 1:1000 for IF (Cat# SA5-10339, RRID:AB_2868386).

**Cell culture.** HeLa cells (type HeLa M) used in this study were a gift from Paul Lehner (University of Cambridge), and were originally published in ref. 54. This is the cell line we have used in several previous DOMs publications[13,15–17,20]. HeLa cells were cultured in Dulbecco's Modified Eagle's Medium (DMEM; Gibco Cat# 31966-021), supplemented with 10% (v/v) foetal bovine serum (FBS; Gibco Cat# 10270106) and 1% (v/v) penicillin-streptomycin solution (Gibco Cat# 15140122). Cells were maintained at 37 °C in a humidified atmosphere of 5% $CO_2$.

**Starvation and BafA treatment.** For starvation, HeLa cells were washed three times with Dulbecco's Phosphate Buffered Saline (PBS) (Gibco Cat# 14190-094) and then incubated for 1 h in Earle's Balanced Salt Solution (EBSS; Sigma-Aldrich Cat# E2888). Where indicated, cells

were incubated in 100 nM Bafilomycin A1 (BafA, Merck, Cat# 19-148) for the stated duration, in full medium (DMEM + 10% FBS) or EBSS (starve + BafA).

**Immunofluorescence microscopy.** For widefield microscopy, HeLa cells were grown onto 13 mm coverslips and fixed in 3% (v/v) formaldehyde in PBS for 20 min at room temperature. Residual aldehyde groups were quenched with 20 mM glycine in PBS for 5 min. Formaldehyde fixed cells were permeabilized with 0.1% (w/v) saponin in PBS for 10 min and blocked in 1% (w/v) BSA/0.01% (w/v) saponin in PBS for 10 min. For labelling of GOLIM4 and TM9SF2, coverslips were instead fixed in 100% ice-cold methanol for 5 min on ice. Methanol fixed cells were washed three times in PBS and blocked in 1% (w/v) BSA in PBS for 10 min. Primary antibody (diluted in BSA block) was added for 1 h at room temperature. Coverslips were washed three times in BSA block and then fluorophore-conjugated secondary antibody (diluted in BSA block) was added for 30 min at room temperature. Coverslips were then washed three times in PBS. Nuclei were stained with DAPI (300 nM in PBS; Thermo Scientific Cat# 62248) for 5 min. Where indicated, coverslips were co-stained with DAPI and Alexa Fluor 488-labelled Phalloidin (330 nM in PBS, Cell Signalling Technology Cat# 8878) for 15 min. Coverslips were washed in PBS, followed by a final wash in ddH2O, before being mounted in ProLong™ Glass Anti-fade Mountant (Invitrogen Cat# P36980).

Microscopy was performed at the Imaging Facility of the Max Planck Institute of Biochemistry, Martinsried, using a Leica DMi8 inverted microscope (Leica Thunder) equipped with a Leica DFC9000 GTC Camera, a 63x/1.47 oil objective (HC PL APO 63x/1.47 OIL) and an iTK LMT200 motorized stage, and controlled by Leica Application Software X (LAS X) version 3.5.5.19976. Cells were selected for imaging using the DAPI channel only in the Navigator software module of LAS X. For the BafA time-course, 10 images were captured for each target protein, per condition and per replicate, using autofocus on the DAPI channel. Images were acquired at a resolution of 2048 × 2048 pixels, 16 bit. For representative images displayed in figures, global linear brightness and contrast changes were performed in ImageJ[55] version 2.1.0 to enhance visualization. However, all quantification of data was performed on the raw unaltered images. For quantification, the dataset was first filtered to remove out-of-focus images.

**Quantification of Golgi protein localization.** Golgi protein localization was quantified by comparing the fluorescence mean intensity of protein targets in the Golgi region (marked by GALNT2) with the mean intensity in the rest of the cell (marked by Phalloidin), from widefield images. The analysis was performed with a workflow implemented in the software CellProfiler, version 4.2.5, complemented with the plugin RunCellpose to allow the use of the Cellpose segmentation algorithm[56] (ver. 2.2) in the pipeline. The pipeline file is available at https://doi.org/10.5281/zenodo.8203066. The position and extension of each cell were determined using a generalist model of Cellpose, providing the Phalloidin and DAPI channels as cytoplasm and nuclear signals, respectively. The region occupied in each cell by the Golgi was determined from the GALNT2 channel, after background subtraction, by minimum cross-entropy thresholding. After illumination correction, the mean fluorescent intensity of the protein of interest was measured for each compartment region, and the ratio of protein intensity between the Golgi and the rest of the cell was calculated for each individual cell. Cells that were not fully included in the field of view were excluded from the analysis. Data were filtered to remove cells where either the nucleus or Golgi was not identified (removing 183 out of 8061 entries). Statistical analyses were performed as described in the statistics and reproducibility section below.

**Generation of label-free dynamic organellar maps.** To avoid sample-related variation during method optimization, we generated three

large-scale replicate maps from HeLa cells, each from 3 × 15 cm dishes at 70–90% confluency, on a single day. Protein samples were digested with LysC and Trypsin (see below). All subsequent peptide clean-ups and peptide fractionations were performed from the same set of digests. For the comparative experiment (Figs. 4 and 5), organellar maps were prepared from control HeLa cells (untreated) and HeLa cells that had been starved for 1 h in the presence of Bafilomycin A1 (100 nM), in triplicate, each from 1 × 15 cm dish at 70–90% confluency. All six maps were generated on the same day.

Cell lysis and subcellular fractionation were performed as reported previously[13,44]. All steps were performed at 4 °C with pre-chilled ice-cold buffers. HeLa cells were washed in PBS (without CaCl$_2$ and MgCl$_2$), incubated in PBS for 5 min, rinsed with hypotonic buffer (25 mM Tris HCl, pH 7.5, 50 mM sucrose, 0.5 mM MgCl$_2$, 0.2 mM EGTA), and immediately incubated in hypotonic buffer for 5 min. Cells were drained, scraped into a total volume of 4 mL of fresh hypotonic lysis buffer and mechanically lysed with 15 strokes of a pre-chilled Dounce homogenizer (7 mL, tight pestle, Kontes Glass Co.). Sucrose was restored to 250 mM with hypertonic sucrose buffer (25 mM Tris HCl, pH 7.5, 2.5 M sucrose, 0.5 mM MgCl$_2$, 0.2 mM EGTA).

All centrifugation steps were performed at 4 °C with the fastest acceleration and deceleration settings. Cell lysates were centrifuged at 1000 × $g$ for 10 min (Multifuge 1 L, Heraeus) to pellet nuclear material and unbroken cells (1 K fraction). Post-nuclear supernatants were transferred to fresh tubes and centrifuged at 3000 × $g$ for 10 min (3 K fraction). Post-3000 × $g$ supernatants were transferred to ultracentrifuge tubes and further sub-fractionated using the Optima™ MAX Ultracentrifuge (Beckman Coulter) with a pre-chilled TLA 110 rotor (Beckman Coulter) by sequential centrifugation steps, each time collecting a protein pellet and transferring the supernatant to a fresh ultracentrifuge tube: 5400 × $g$ for 15 min (6 K fraction), 12,200 × $g$ for 20 min (12 K fraction), 24,000 × $g$ for 20 min (24 K fraction), and 78,400 × $g$ for 30 min (80 K fraction). All pellets were resuspended in 1×SDS buffer (2.5% SDS, 50 mM Tris HCl, pH 8.1). The supernatant obtained after the final centrifugation step (cytosolic fraction) was mixed at a 4:1 ratio with 5×SDS buffer (12.5% SDS, 50 mM Tris HCl, pH 8.1). Samples were heated at 72 °C for 5 min and sonicated using a Bioruptor (Diagenode Inc) with fifteen 30 s on/off cycles at maximum intensity. Fully solubilized samples were stored at −80 °C. Protein concentrations were determined using the Thermo Scientific™ Pierce™ BCA (bicinchoninic acid) Protein Assay Kit (Thermo Scientific™ Cat# 23225). Following concentration determination, DTT (Sigma-Aldrich Cat# D0632-25G) was added to a final concentration of 1 mM before preparing the samples for mass spectrometry.

**Sample preparation for mass spectrometry.** For in-solution digestion, protein was precipitated by the addition of five volumes of ice-cold acetone, incubated at −20 °C overnight and pelleted by centrifugation at 10,000 × $g$ (Centrifuge 5418 R, Eppendorf) for 5 min at 4 °C. All subsequent steps were performed at room temperature. Precipitated protein pellets were drained, air-dried for 5 min, resuspended thoroughly in urea buffer (8 M urea, 50 mM Tris HCl, pH 8.1, freshly added 1 mM DTT), and incubated for 15 min. Sulfhydryl groups were alkylated by the addition of 5 mM iodoacetamide for 1 h in the dark. Proteins were enzymatically predigested by the addition of LysC (1 μg per 50 μg of protein; Wako Cat# 129-02541) for overnight incubation. Predigests were then diluted four-fold with 50 mM Tris, pH 8.1 (final urea concentration = 2 M) before addition of trypsin (1 μg per 50 μg of protein; Sigma-Aldrich Cat# T6567) for a 3 h incubation. The reaction was stopped by the addition of 1% trifluoroacetic acid (TFA, final pH <3). Samples were incubated on ice for 10 min and spun at 10,000 × $g$ for 5 min at 4 °C. Supernatants were transferred to fresh tubes for peptide storage at −20 °C.

Peptides were purified either by solid-phase extraction with poly(styrenedivinylbenzene) reverse-phase sulfonate (SDB-RPS), as

previously described[27], or by LC trapping using commercially available C18 StageTips (EvoTips Cat# EV2001) of the Evosep System, according to the manufacturer's instructions. In brief, EvoTips were activated by wetting the C18 material in 1-propanol, washed with Evosep buffer B (0.1% [v/v] formic acid in acetonitrile), and wetted in 1-propanol again for 5 min. Soaked tips were washed with Evosep buffer A (0.1% [v/v] formic acid), then with 0.2 % formic acid and then loaded with 200 ng acidified peptide sample. EvoTips were washed with Evosep buffer A, and finally loaded with Evosep buffer A and stored at 4 °C until analysis by mass spectrometry. Peptides purified via the SDB-RPS approach were dried at 45 °C in a centrifugal vacuum concentrator (Concentrator 5301, Eppendorf), resuspended in buffer A* (0.1 % [v/v] TFA, 2% [v/v] acetonitrile), and stored at −20 °C until analysis by mass spectrometry.

For deep measurements and the DDA library for the 100 min gradient, peptides were triple-fractionated on SDB-RPS StageTips[27]. StageTips were washed with 100% acetonitrile, equilibrated with StageTip equilibration buffer (30% [v/v] methanol, 1% [v/v] TFA), and washed with 0.2% (v/v) TFA. 20 μg of peptides in 1% TFA were loaded onto activated stage-tips, washed with isopropanol, and then twice with 0.2% (v/v) TFA. Peptides were eluted in three consecutive fractions by applying a step gradient of increasing acetonitrile concentrations: 20 μL SDB-RPS-1 (100 mM ammonium formate, 40% [v/v] acetonitrile, 0.5% [v/v] formic acid), then 20 μL SDB-RPS-2 (150 mM ammonium formate, 60% [v/v] acetonitrile, 0.5% [v/v] formic acid), then 30 μL SDB-RPS-3 (5% [v/v] NH$_4$OH, 80% [v/v] acetonitrile). For the DDA libraries for the 21 and 44 min gradients, peptides were fractionated into a final eight fractions using a Pierce High pH reversed-Phase Peptide Fractionation Kit (Thermo Fisher Scientific, 84868), according to the manufacturer's instructions.

**Mass spectrometric analysis.** All measurements were performed on a Thermo Exploris 480 mass spectrometer, with minimal chromatography column changes. Several MS setups and strategies were tested, most importantly data independent vs data-dependent acquisition. The effect of gradient length on map quality was evaluated for 21, 44 and 100 min gradients, for both triply SDB-RPS fractionated[27] and unfractionated samples.

Nanoflow reversed-phase chromatography was performed using either the Evosep One (Evosep Biosystems) or the EASY-nLC 1200 ultra-high-pressure system coupled online to an Orbitrap Exploris 480 instrument via a nano-electrospray ion source (all Thermo Fisher Scientific). On the EASY-nLC 1200 system a binary buffer system with the mobile phases A (0.1% [v/v] formic acid) and B (80% acetonitrile, 0.1% [v/v] formic acid) was employed. Peptides were separated in 100 min at a constant flow rate of 300 nL/min on a 50 cm × 75 μm (i.d.) column with a laser-pulled emitter tip, packed in-house with ReproSil-Pur C18-AQ 1.9 μm silica beads (Dr. Maisch GmbH). The column was operated at 60 °C using an in-house manufactured oven. In total, 300 ng of purified peptides in Buffer A* were loaded onto the column in Buffer A and eluted using a linear 84 min gradient of Buffer B from 5 to 30%, followed by an increase to 60% B in 8 min, a further increase to 95% B in 4 min, a constant phase at 95% B for 4 min, followed by washout—a decrease to 5% B in 5 min and a constant phase at 5% B for 5 min—before re-equilibration. On the Evosep One LC system a binary buffer system with the mobile phases A (0.1% [v/v] formic acid) and B (0.1% [v/v] formic acid in acetonitrile) was used. Peptides were separated in 21 min at a flow rate of 1.0 μL/min on an 8 cm column (with a throughput of 60 samples per day [SPD]) or 44 min at a flow rate of 0.5 μL/min on a 15 cm column (with a throughput of 30 SPD), using in-house packed columns and standard pre-programmed gradients. The 15 cm in-house packed column was operated at 60 °C using an in-house manufactured oven.

For DDA, the Orbitrap Exploris 480 mass spectrometer run by Xcalibur (v.4.4, Thermo Fisher) was operated in top 15 scan mode

(DDA) with a full scan range of 300– 1650 Th when coupled to the EASY-nLC 1200 system (100 min gradient). Survey scans were acquired at 60,000 resolution with an automatic gain control (AGC) target of $3 \times 10^6$ charges and a maximum ion injection time of 25 ms. The selected precursor ions were isolated in a window of 1.4 Th, fragmented by higher-energy collisional dissociation (HCD) with normalized collision energies of 30. Fragment scans were performed at 15,000 resolution, with a maximum injection time of 28 ms, an AGC target of $1 \times 10^5$ charges, and a precursor dynamic exclusion for 30 s. Acquisition schemes for the data-independent acquisition (DIA) scan mode used here were described previously[57,58], but were optimized and tailored for the Dynamic Organellar Maps approach. In brief, the DIA method for the 100 min gradient consisted of one survey scan that was followed by 33 variably sized MS2 windows (17–161 Th) in one cycle, resulting in a cycle time of 2.5 s. Survey scans were acquired at 120,000 resolution with an AGC target of $3 \times 10^6$ charges and a maximum injection time of 60 ms covering a $m/z$ range of 350–1400. MS2 scans were acquired at 30,000 resolution with an Xcalibur-automated maximum injection time, covering a $m/z$ range of 332 (lower boundary of the first window) to 1570 (upper boundary of the 33rd window). The DIA method for the 44 min and 21 min gradient consisted of one survey scan that was followed by 35 equally sized MS2 windows (19.2 Th with 1 Th overlap) in one cycle, resulting in a cycle time of 1.5 s. Survey scans were acquired at 120,000 resolution with an AGC target of $3 \times 10^6$ charges and a maximum injection time of 45 ms, covering a $m/z$ range of 350–1400. MS2 scans were acquired at 15,000 resolution with a maximum injection time of 22 ms, covering a $m/z$ range of 361–1033.

### Raw data analysis

For peptide and protein identification, MS raw data were imported into MaxQuant version 2.1.3[34]. Unless otherwise stated, default parameters were used for all settings. The MS2 spectra were searched against the SwissProt entries contained in the UniProt human reference proteome FASTA database (UP000005640_9606, 42,418 entries). Spectral libraries were constructed using DDA raw data of fractionated subcellular samples of the same organellar maps that were used for the data acquired in DIA mode.

**Spectral library generation and DDA analysis.** For spectral libraries and the DDA analyses, DDA raw files were processed in MaxQuant[32,59] employing the Andromeda search engine[60]. For accurate label-free quantification, the 'MaxLFQ algorithm'[22] was enabled with LFQ minimum ratio count of 1 and the match-between-runs feature was enabled to match between equivalent subcellular fractions of replicates. Each spectral library was assembled from 21 samples or 56 samples (six subcellular fractions as used for mapping, plus cytosol, each fractionated at the peptide level threefold or eightfold as described above). A dedicated library was generated for each LC gradient length (100 min, 156.7 K peptides; 44 min, 88.5 K peptides; 21 min, 58.7 K peptides).

**DIA analysis.** DIA raw files were processed via MaxDIA[34], which is embedded into the MaxQuant software environment, using default settings except for using a minimum LFQ ratio count of 1 and disabling large ratio stabilization. For both the discovery and library DIA approaches, spectral libraries of peptides were provided in the form of 'peptides', 'evidence', and 'msms' files. Whereas for the library approach these files were obtained from MaxQuant DDA searches, for the discovery approach an in silico predicted library for all human peptides with up to 1 missed cleavage was used. The prediction had previously been generated using the DeepMass:Prism tool[35]. The provided library was filtered to contain only Swiss-Prot entries, using a python script (github.com/cox-labs/DIAtools/tree/main/Misc/FilterAdditional).

### Data analysis using the DOM-ABC web app

The intra- and inter-experimental quality of the dynamic organellar maps were evaluated to assess the performance of different combinations of MS methods, LC-MS setups, and processing strategies. To enable the visual exploration, quality assessment and analysis of spatial proteomics data, we developed DOM-ABC, a web-based app (https://domabc.bornerlab.org). The workflow is entirely based on the Python scripting language and uses several external libraries as documented on github (https://github.com/JuliaS92/Spatial ProteomicsQC). DOM-ABC performs customizable data filtering, normalization, and graphical representation. Various analysis tools allow detailed exploration of data, map quality, reproducibility, and resolution, as well as performance of localization prediction and translocation analyses. All results can be downloaded as support vector graphics, formatted tables and as comprehensive.json files for custom analysis. Importantly, several maps can be compared in parallel. Settings defining the downstream analysis can be downloaded to ensure reproducible analysis. All map analyses shown in the paper were performed using six-point profiles, i.e., protein abundance across 1 K, 3 K, 6 K, 12 K, 24 K and 80 K fractions.

**Data filtering.** The primary output from MaxQuant or Spectronaut, or any tabular data with profiling-based protein quantifications, can be loaded into DOM-ABC. Per default settings for the MaxQuant output, reverse hits, contaminants and proteins only identified by sites are removed. Further filtering is then performed at the level of individual maps and tailored to each quantification strategy, to obtain datasets with high-quality measurements. All of these steps are defined with default settings, but can be parameterized differently or individually disenabled through the graphical user interface.

For SILAC maps, SILAC ratios are retained if they are based on more than two quantification events, or on two quantification events where the ratio variability was below 30%. For each fraction, SILAC ratios are normalized by dividing by the median ratio for the fraction. Only proteins with complete profiles are retained, i.e., a valid SILAC ratio in each subcellular fraction. SILAC ratios are inverted (assuming that the reference fraction is SILAC heavy[13]) and profiles for each protein are 0-1 normalized, as follows. For each protein, the ratios are summed across the six fractions. Each ratio is then divided by the summed total for the protein. For LFQ maps, intensities are already globally normalized, hence no further normalization is required. Two stringency filters are applied: First, only profiles with LFQ intensities in at least four consecutive fractions are considered. Second, profiles are rejected if their mean MS/MS count per subcellular fraction is less than two. Then, (0-1) normalization of each profile is performed by summing the intensities across all fractions and dividing each intensity by the summed total for the profile. Filtered datasets are annotated based on a predefined set of 1076 organellar marker proteins covering 12 subcellular localizations/organelles[13]. These default settings were used for all datasets in this study.

**Protein group alignment.** To compare quantifications from different raw data processing runs, protein groups need to be aligned. We implemented a strategy in which we prioritize matching of single-gene locus protein groups with complete coverage across experiments, over matching of (rare) multi-gene locus groups and groups with incomplete coverage. First, single-gene locus protein groups are temporarily reduced to the canonical id (if present), and otherwise to the first listed isoform id. Protein groups that can be found in all compared runs are then re-labelled to the reduced protein group id and flagged as 'primary id' matches. Typically, this is the case for the majority of proteins. Second, we maximize overlap for the remaining multi-gene locus groups, and single-gene locus groups with incomplete coverage, at the cost of making less exact matches. Starting with the largest multi-gene locus group across all experiments, we match

this with its largest remaining subset in each other experiment, removing them from the pool of available groups. This is repeated until no multi-gene locus groups remain. These matches are re-labelled with all protein ids contained within the group, and either flagged as 'multiple genes', if all matched protein groups cover identical loci across experiments, or as 'gene level conflict', if different loci were covered in different experiments. At the end of this procedure only single-gene locus groups with incomplete coverage remain to be aligned. These are reduced in the same way as the single-gene locus full coverage groups and are also flagged as 'primary id' matches. To ensure full traceability of the original protein grouping in each of the compared search engine runs, an id mapping table is stored and available for download.

**Assessment of proteomic depth.** Proteomic depth is assessed by counting protein groups that are either identified in one or all replicates, and by counting proteins that are fully profiled (i.e., passing all quality filters) in one or all replicates. Venn diagrams and upset plots are provided to evaluate overlap of proteins.

**Principal component analysis.** For graphical map representation, filtered and 0-1 normalized data from all experiments compared were centered and scaled to unit variance in each fraction. To do this, mean intensities were calculated for each fraction and subtracted from each individual intensity of the corresponding fraction (centering). Subsequently, each intensity was divided by the intensity standard deviation of this fraction (scaling to unit variance). Maps were then jointly subjected to principal component analysis (PCA) to achieve dimensionality reduction. For each map, the first three principal components were calculated via Python's scikit-learn library[61]. For HeLa cells scores plots of PCs 1 and 3 usually provide the best visual resolution of postnuclear clusters, as PC 2 is dominated by the nuclear 1 K fraction. The tool also provides the option to calculate further principal components, and shows elbow and loading plots together with the 2D or 3D PC projections. When interpreting distances in PCA space as done in Fig. 5, the axes should be scaled according to variance, which can be toggled in DOM-ABC.

**Profile scatter within stable complexes (intra-map scatter).** The subunits of a stable protein complex have identical subcellular distributions, and should therefore have very similar abundance profiles. Observed deviations are mostly caused by MS measurement noise, and intra-complex protein scatter thus reflects within-map quantification precision. We curated a dataset of around 30 well-characterized protein complexes with at least five subunits (e.g., 20S core proteasome, CCT, COPI). Within DOM-ABC, starting with the filtered and 0-1 normalized data, profiles that belong to a specified protein cluster are extracted and filtered to leave only proteins that were measured across all compared maps and experiments. By default, only complexes with full coverage data for at least five proteins are analyzed. Subsequently, the absolute distance (Manhattan Distance) of each subunit profile to the complex median profile is calculated. Smaller distances suggest more precise quantification. We observed that the baseline scatter varies somewhat between complexes, and therefore normalize the acquired distances for comparison of experiments. To this end, for each protein all distances are divided by the median distance across all experiments. These values follow a lognormal distribution and are aggregated per complex by median calculation. In DOM-ABC it is possible to also normalize to a specific experiment if that is preferred. We recommend that the overall assessment should be based on at least ten different complexes.

**Inter-maps profile reproducibility (inter-maps scatter).** To evaluate the reproducibility of 0-1 normalized profiles, the inter-profile scatter across replicates is calculated. For each protein, the absolute distance (Manhattan Distance) of each replicate profile to the mean profile from all replicates is calculated; these distances are averaged to obtain this protein's profile scatter. Global profile scatter is then plotted as a density function for proteins common to all compared maps. As an additional output, the distribution for all profiled proteins can be displayed, regardless of overlap with the other examined maps. The greater the proportion of proteins with low scatter, the better the between-maps reproducibility. As a numeric readout, the scatter at a specified quantile of each distribution can be displayed.

**Support vector machine analysis.** To further evaluate the performance of organellar maps, their power to predict protein localization was assessed using quality-filtered, (0-1) normalized data with full replicate coverage. For supervised classification a set of marker proteins covering 11 subcellular localizations was used as a means to assign all other proteins to organellar clusters by SVMs; the previously defined ER_high_curvature cluster[13] was removed in this study due to the low number of marker proteins in the depth-limited short-gradient datasets. As far as practicable (see figure legends), only markers present in all compared datasets were included, and identical SVM parameters were used. Machine learning was done using the SVM module of DOM-ABC, which is based on functionality provided in scikit-learn[61]. The SVM module enables selection of marker classes, definition of a hold-out test set, automated hyper parameter optimization and finally SVM training and prediction. During training the parameters C and gamma of the radial basis function are optimized via an iterative grid-search, employing fivefold cross-validation. At each grid point SVMs are run on each loaded dataset and the optimum for the summed accuracy across datasets is found. For prediction the SVM is fit using the training set and fivefold cross-validation is used for converting the raw SVM scores into probabilities. Proteins are then assigned to the best fitting organelle model and divided into confidence classes, based on the probability: >0.95 very high confidence; >0.8, high confidence; >0.65, medium confidence; >0.4, low confidence; <0.4, best guess assignment. For the hold-out test set a misclassification matrix (Supplementary Data 4) is derived to calculate the global marker prediction recall (proportion of correctly predicted to the total number of markers), the organelle specific recall (proportion of markers correctly assigned to the cluster), and the organelle specific precision (ratio of markers correctly assigned to the number of all markers assigned to the cluster). The harmonic mean of recall and precision, the F1 score, was used as the primary readout for SVM performance. To assess the variability of these scores, the test set is sub-sampled 20 times (class stratified 75%) and the mean scores ± standard deviation are reported. For scores averaged across organelles, the error bar also represents the deviation across these samples and not the deviation across organelles. DOM-ABC is also able to accept misclassification matrices from external tools like Perseus[52] and MetaMass[50] as input.

**Protein subcellular localization shift analysis.** We compared organellar maps from untreated HeLa cells and from HeLa cells that had been starved in the presence of BafA for 1 h. DOM-ABC contains a module that implements our previously established MR analysis[13,44] to identify proteins with significant subcellular localization shifts. Replicates were numbered 1–3. Using the default settings of the DOM-ABC, 5672 normalized high-quality profiles were obtained across all six maps. To ensure that the next analysis steps are not affected by misquantified profiles, we further removed proteins where the smallest cosine correlation between any replicates within one condition was <0.9. This left 4475 proteins in the set for DIA data. Next, delta profiles were calculated within each cognate pair of untreated and treated maps, by subtracting the 0-1 normalized profiles from a treated map from the profiles of its matching control map. For each of the three obtained sets of delta profiles, a multidimensional outlier test based on the robust Mahalanobis distance was performed[62], using functions

from scikit-learn[61] (proportion of data used = 0.75, median of 31 iterations with different random states). These distances follow a chi2 distribution and obtained p-values reflect the probability of observing a given profile shift (or a greater shift) by chance. For each protein, the three *p*-values from the three replicates where then combined using the Fisher method. The combined *p*-values where corrected for multiple testing using the Benjamini-Hochberg false-discovery-rate (FDR) approach, and −log(10) transformed to obtain a Movement score (M score). We chose an FDR of 0.05 as our initial stringent cut-off, corresponding to an M-score of 1.3. We also included the requirement that at least two of the replicate p-values need to be <0.1, to filter out proteins with single very small *p*-values in only one replicate. As a further stringency filter, we also calculated a reproducibility score (R score). For each protein, the Pearson correlation of all pairs of delta profiles was calculated (Rep 1 vs 2, 1 vs 3, 2 vs 3) and the median correlation was designated as the R score. Profile shifts with $M > 1.3$ and $R > 0.75$ were considered significant and reproducible.

### DOM-ABC 1-min quick start guide
This guide will allow you to test the DOM-ABC tool with data generated in this study.

1. Go to webpage https://domabc.bornerlab.org.
2. Click on the big green button 'Benchmark multiple experiments'.
3. From the 'Add reference set' drop-down menu (top right corner), select 'HeLa 1×100 min libraryDIA'. Click the 'Load' button.
4. Repeat Step 3., to load the 'HeLa 1×100 min DDA' file.
   You have now loaded two different sets of maps.
5. Click the big green button 'Align and analyse selected datasets'. This may take a moment—the program will update on progress and tell you when it's finished (bottom right corner).
6. Scroll down, and select the 'Overview', 'PCA maps', 'Depth and coverage',... tabs to view the different analyses.

The sample 'reference sets'.json files are already integrated into the DOM-ABC tool. To analyse the complete datasets generated in this study, upload the .json files provided as Supplementary Data 5. To do so, in step 3, click the 'Browse' button, select a .json file, and upload.

To configure your own analysis of profiling data, go to the start page and follow the instructions.

### Downstream analysis of translocating proteins
**Clustering and enrichment analysis of moving proteins.** To group and label the detected moving proteins, hierarchical clustering and annotation enrichment were used, applying previously published python code[63], that largely relies on functions from the widely used scipy library. For all 164 outliers, delta profiles were first subjected to hierarchical clustering. Variable parameters are the distance metric, linkage method and distance threshold. Complete linkage was chosen, because it yielded highly similar dendrograms independent of the distance metric. Pearson correlation as distance metric yielded the clearest and visually most easily comprehensible clustering. The cutoff was chosen such that all visually apparent groups of shifts were complete and well separated at the same time, which yielded the 7 clusters reported in Fig. 4 and Supplementary Data 2. All 164 proteins were then annotated with GO terms for cellular compartment, biological function and molecular function, as well as protein families, all downloaded from Uniprot (30.9.2021). A fisher's exact test was applied for each annotation term, for each cluster against the full list of transitioning proteins. Resulting *p*-values were corrected for multiple hypotheses using Benjamini-Hochberg correction and a cutoff of 10% FDR was applied. Two clusters did not yield any enriched terms at this cutoff and thus remained unlabelled.

**Sensitive detection of cycling proteins by clustering.** To find other moving proteins that have a similar direction, but lower shift

magnitudes than the proteins detected by the M-R analysis, we again used hierarchical clustering. As we were looking specifically for Golgi localized proteins following a similar trajectory as the 10 Golgi proteins among our 164 hits, we first curated a list of Golgi proteins. This included the 10 hits, Golgi marker proteins, proteins classified as Golgi proteins by SVM predictions based on the non-treated data set, and proteins classified as Golgi proteins in[13]. From this list we only included lumenal and transmembrane proteins. The full pairwise correlation matrix between these proteins was calculated and used as input for hierarchical clustering, as described above, but with Euclidean distance as the metric for clustering. Using the correlation matrix instead of the delta profiles as input retains the information from all pairwise comparisons and clusters proteins by their phenotype similarity to all other proteins in the set, rather than the raw profiles, which is more robust.

To categorize endosomal shift magnitudes, we calculated endosomal shifts, i.e., the difference between the correlations with the mean endosomal marker profiles before and after treatment, for all 5672 proteins. These differences follow a fairly symmetrical and in the central part roughly normal distribution around zero (Supplementary Fig. 6B). We robustly z-scored shifts using the median of −0.007 and a robustly estimated standard deviation of 0.141 (based on the median absolute deviation from the median). z-scores where then used to classify shift magnitudes as follows: $z < 0.5$, no relevant shift; $z = (0.5-1)$, very small; $z = (1-2)$, small; $z = (2-3)$, medium; $z = (3-4)$, large; $z > 4$, very large. Further manual inspection of profile shifts (Fig. 4d; interactive Supplementary Data 3) allowed us to label very large shifts as complete endosome transitions, and smaller shifts as partial endosome transitions.

### Statistics and reproducibility
All dynamic organellar maps were prepared in biological triplicates and reproducibly showed similar results as outlined in the "Results" section.

Images show in Fig. 6 are representative of at least 9 images per condition from two biological replicates (independent starvation/BafA treatments performed on separate days), with immunofluorescence labelling and microscopy performed independently for each replicate. Images shown in Fig. 7 are representative of at least 16 images per condition from two biological replicates (independent BafA treatments performed on separate days), with immunofluorescence labelling and microscopy performed independently for each replicate. Images shown in Supplementary Fig. 6C are representative of 3 images per condition from one biological replicate.

Statistical analyses of imaging data were performed in GraphPad Prism version 9.4.1 for Windows. The Golgi protein localization data shown in Fig. 7 were analyzed using a Kruskal−Wallis test with Dunn's Multiple Comparisons post-test for comparisons to the 0 h timepoint. The number of cells analyzed varied between samples; $n$ = a minimum of 125 cells per condition examined over two independent experiments, except for the 6 h timepoint for TM9SF2 for which $n$ = 125 cells examined over one independent experiment. All individual $n$ values and resulting $p$ values are reported in Supplementary Table 1. Replicate data were combined for plotting and statistical analyses, but separate analyses of each individual replicate delivered consistent results.

### Reporting summary
Further information on research design is available in the Nature Portfolio Reporting Summary linked to this article.

## Data availability
The mass spectrometry proteomics data generated in this study have been deposited to the ProteomeXchange Consortium [http://proteomecentral.proteomexchange.org] via the PRIDE partner repository under the accession numbers PXD034962 (DDA data),

PXD034971 (DIA data acquired with the optimized MS methods), and PXD034969 (raw data for the comparative experiment). The imaging data generated in this study have been deposited on Zenodo and are available at https://doi.org/10.5281/zenodo.8197844. All DOM-ABC benchmarks in this study can be replicated by upload of the .json files, provided in Supplementary Data 5, to https://domabc.bornerlab.org. See DOM-ABC 1-min Quick Start Guide above. Source data are provided with this paper.

## Code availability

The web app DOM-ABC introduced in this study is available at https://domabc.bornerlab.org, and the source code at https://github.com/JuliaS92/SpatialProteomicsQC and version 1.0.0 was deposited at https://doi.org/10.5281/zenodo.8219481. The image analysis script used in this study is available at https://doi.org/10.5281/zenodo.8203066.

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

## Acknowledgements

J.S., V.A., P.S., and G.B. were supported by the Max-Planck Society for Advancement of Science. A.D. received funding from the European Union's Horizon 2020 research and innovation programme under the Marie Sklodowska-Curie grant agreement no. 896725 and a Humboldt Research Fellowship from the Alexander von Humboldt Foundation. We wish to thank Matthias Mann for his continued generous support. We also want to thank members of the department for Proteomics and Signal Transduction for fruitful discussions and providing starting points for the DIA method optimization: Isabell Bludau, Sophia Steigerwald, Maximilian Zwiebel, Marvin Thielert, Patricia Skowronek, Jakob Bader and Florian Meier. We also thank Jürgen Cox for providing us with MaxQuant 2.0 prior to its release. We thank the MPIB Imaging Facility for outstanding technical support, in particular Giovanni Cardone for his advice and implementation of image analysis pipelines. We are very grateful to Igor Paron, Tim Heymann and the column team for outstanding technical support. The illustration of the cell culture dish in Fig. 1 was previously published[15]. We would like to thank Lisa Schweizer and Sophia Steigerwald for their help with drawing the HPLCs and the mass spectrometer in Fig. 1.

## Author contributions

G.B., J.S. and V.A. devised the study; J.S. conceptualized and designed DOM-ABC; J.S. and V.A. implemented DOM-ABC; J.S., V.A. and G.B. analyzed the MS data; V.A. and A.D. performed the organellar mapping experiments; A.D. designed, performed and analyzed the imaging experiments; A.D. and V.A. performed the BafA timecourse and immunofluorescence labelling; V.A. ran MS acquisitions and raw data analyses; P.S. contributed to raw data processing in MaxQuant and code review; J.S., A.D., V.A. and G.B. wrote the original drafts of the manuscript (initial and revised versions); all authors contributed to reviewing and editing of the final manuscript; G.B. supervised the project.

## Funding

## Competing interests

The authors declare no competing interests.
