## [Peer Review File · Nature Communications]

REVIEWER COMMENTS

Reviewer #1 (Remarks to the Author):

Schessner et al. describe a novel method to create Dynamic Organellar Maps (DOMs) in human cells by combining fractionation and DIA acquisition and claim a substantial improvement in coverage and timing. Although this is potentially interesting for the proteomics community, I do not see the novelty here, rather a much-improved version of their previous published method. Even in the introduction (starting from the second paragraph and until the end) the authors give very technical details of the previous method which are not necessarily useful for the broader readership of Nat Communications which also include experts in cell biology, biochemistry, cell signalling, cancer, etc. The first part of results is again very technical and regards the development of software for data analysis which would be more suitable for a specialistic proteomic journal. Also evaluating the performance of LC is more suitable for proteomics journal than for the broader readership of this journal. Studying the effect of starvation and bafilomycin A1 treatment is interesting but the authors do not explain why they want to use these treatments (which is the big picture, the big biological problem to solve here?) or why they developed this method to study autophagy and changes in protein localization. However, the finding that “GLG1 showed complete transition from Golgi to a punctate endosomal pattern, TGOLN2 underwent a partial translocation, and GALNT2 retained its Golgi pattern (Fig. 5G)” is potentially very interesting. Any functional follow up? What happens whether this translocation does not happen? Validating this novel idea would make the paper novel.

Reviewer #3 (Remarks to the Author):

Schessner et al benchmark DIA-MS approaches for DOM measurements, present a web implemented tool for QC analysis of DOM experiments and perform DOM to characterize Golgi proteins that cycle through endosomes. This study is relevant to the field of spatial proteomics and the DOM method has the potential to be broadly applied due to its simplicity. The described method optimizations provide significant advantages, and the web application will be of value for proteomic researchers interested in using DOM. The endosome-Golgi experiment is well performed, and the authors followed up and validated several hits.

Major Comments:

1. For the MS method optimization and comparisons (Figure S1) the author should provide additional QC measures. What was the protein coverage, peptide/protein CVs, MS2 measurement points per peak and peptide measurement missingness for the different methods?

2. The paper includes a vast number of non-trivial scores that need to be understood to interpret the figures and text. The scores are mostly described in the method sections. It would enhance the clarity of the manuscript if the authors could introduce the meaning of important scores when they first mention them in the text. For example: intra-complex scatter, inter-replica scatter, F1 score, R score, M score

3. It would improve the usability of the QC web application if there were some description (eg figure legends) for the individual plots and how different outcomes should be interpreted.

4. It is unclear how low, medium, and high-confidence localization prediction are determined/defined.

5. Regarding the stacked intra-complex scatter plots. It would be good if the authors provide some additional analysis/explanations for those. For example, Figure 2E: As the authors state library DIA leads to the overall lowest combined intra-scatter. However, for several complexes (eg mitochondrial ribosomes, MCM complex) library DIA has noticeably higher intra-complex scatter. Is there an explanation for this? Maybe it has something to do with differences in protein coverage?

6. Regarding their PCA plots (Fig 2, 4, 5). How much variance is typically captured by PC1, 2 and 3? Would be good to report, so that users have a reference. Why do the authors always show PC3 instead of 2? What is contained in PC2? Fig 5 C-F: are these based on the combined PCA plots from Fig4? Why do they show PC2 here instead of PC3?

7. It would be very useful if the web-tool included a way to identify differentially localized proteins and generate M-R plots. The described analysis to get to the differentially localized proteins and M-R plots is not very streamlined (eg including Perseus analysis) and might be a real bottleneck for people interested in using the DOM method. However, I understand if implementation of this analysis in the webtool is out of the scope for this manuscript.

Minor Comments

8. PCA from Figure 4 vs Figure 2. Do I understand correctly that PCA in Figure 2 shows all proteins whereas PCA in Figure 4 only shows marker proteins? It would be good to plot the PCA in Figure 4 the same way as Figure 2 so that comparisons between Figures 2 and 4 are possible.

9. Unclear meaning of: “enhanced tightness of organellar clusters was apparent in the DIA maps” – page 4.

10. Supplementary Fig. 2D is not linked or described anywhere.

11. FDR = 5% should not be described as “high stringency cut-off” – page 7.

12. Use “m/z” instead of “Th” (method section).

The Borner lab present an updated version of the DOM workflow, combining it with DIA-MS. The manuscript suggests that their approach is more applicable and more straightforward to implement. I can see that this method makes their approach more widely accessible. Their updated workflow is accompanied by new software tool for quality control of their workflows. I commend them for making it open source as too much proteomics software is not and holding back the community. I get a positive impression from the manuscript and can see it as a valuable scholarly contribution. The authors attention to detail is very high, which makes the manuscript enjoyable to read. I remain to be convinced on several point in the manuscript, however, and I propose that the authors can address them to make their manuscript more solid (and in some parts mathematically correct). I hope the authors find my comments useful to improve their manuscript.

Overall comments:

The manuscript describes an approach that contains elements that are similar to already published and established methods. This would not be much of an issue provided that this work was more carefully positioned with respect to these methods in terms of comparisons and inclusive citations. I find some parts of the manuscript somewhat oversold and I would hope to see a more realistic description of what the method can offer in the final version of the manuscript.

In particular, I have two points that may require some significant re-thinking on the part of the authors that are more high-level concerns rather than points about specific arguments. The manuscript would benefit from addressing these points.

Firstly, the method is somewhat similar to the approach published earlier in the year by the Olsen/Lund-Johnson lab. I appreciate that publication uses chemical fractionation, but other aspects are quite similar. Furthermore, the depth of application and demonstration in that paper is considerably higher (phosphoproteomics and non-cell line samples) and from reading both manuscript methodological details, the Olsen/Lund-Johnson method seems to require less starting material. There is a very cursory comparison in the discussion, and I think it would be better if readers could see much more clearly the differences between the approaches. This could include MS-time, instrument/reagents needed, applicability, reproducibility, resolution. I think a more quantitative comparison is needed.

Secondly, the paper is somewhat a 3-1, a software tool, an improved DOM method and application to lysosomal pH disruption. The paper might be better split into three papers as this paper is quite difficult to follow what the contributions are. Currently, none of the 3 separate parts are that convincing for anything other than bioinformatics/proteomics journals. The software tool doesn't appear to offer anything other than simple metrics and things already implemented and easily useable elsewhere. Whilst there is a clear improvement by using DIA, it is not much of a conceptual advance to apply DIA to DOM and can be considered somewhat incremental. The authors suggestion of DIA SILAC DOMs or demonstrating that their approach finds new proteins in their lysosomal pH application would have greatly improved the manuscript. For example, it's not clear that their new-found proteins in their application could not have been found using their original protocol. It's not clear that just finding more proteins brings any benefit to the methodology. The application is somewhat under-developed and could be complemented by a number of assays to substantiate their claims about translation/autophagy changes. For example, I'm not at all convinced that ribosomal protein shifting towards to the highest speed fraction is consistent with "starvation-induced reduction of translational activity".

Major comments:

The authors argue this is the simplest and easiest method to implement amongst the methods available. However, I couldn't really find any details about the other methods other than a selective set of cursory citations. Could the authors contextualise their method within the literature?

The same is true of their QC tool, could the authors contextualise this tool amongst those already available in the spatial proteomics community? What is offered beyond MetaMass and QSep?

The authors argue for improved reproducibility which I can mostly see from their analysis. The authors are missing simple x-y plots of the protein/peptide quantitation values at each centrifugation speed, where the data are compared one replicate against the other. Spearman correlation accompanying these would also add value. I can also see that a direct comparison to the Olsen data would be valuable here.

Assessing the quality of the data through the F1-score is probably not the best idea. The issue is that it is highly susceptible to 1) the markers chosen 2) the strong class imbalance 3) machine learning methods chosen. The authors can remedy these points by considering the following analysis. The authors can change the marker set used, either by sub-sampling, considering different marker sets from the compartments database and/or using markers from different labs. The F1 scores are also presented without error bars, this can be improved by repeated sub-sampling of the hold-out test set used for prediction and comparing a distribution of scores. The strong class imbalance can be tackled with several analysis, a) being more explicit about the false positives and false negatives (I know the F1 score combines these but it's good to be explicit) b) show the distribution of scores over the different subcellular niches and c) plot precision-recall curves and compute the area under the curve. Also demonstrating the results hold for say the k-NN algorithm and the random forest method would show that it's not simply over-optimisation to a particular ML algorithm but rather a generally improved methodology.

I have some concerns with what the PCA plots are showing. Firstly, they are displayed incorrectly: they should show the variance explained in each component – otherwise they are uninterpretable, and the ratio of the axis should be comparable to the ratio of the variance explain. To be explicit the height of a PCA plot cannot ever be greater than the width. I'm also concerned that the authors suggest the best dimension to plot are 1 and 3. Strong experimental design would suggest that desired quantity (separation of subcellular compartments) dominates the variance in the data. However, the authors are suggesting that this is not true and there is some other reason for the scatter in PC2. Why is this the case? Is there a contamination driving the variability in this dimension? It would be good to investigate what is happening here. Finally, the Golgi proteins look quite different in each PCA plot (Fig. 2), I fail to find the nuclear compartment (it is just not labelled?) and the DDA data looks like it has fewer outliers.

I have to say I stared at Fig. 2E for quite some time to understand it and still do not really know what I'm expecting to see. I think it probably that the information density and colours that are making this plot difficult to read. Fig. 3C has the same problem. Is there a better way to display this information?

It's hard to assess the scale of the differences in Fig. 2B. What do these changes in F1 score mean in practice and what is the associated error with these values?

Figure 4 has some issues. Could the PCA plots be fixed as suggest above? I think panels B and C do not add much so could be sent to the supplementary materials. I am confused by panel D, I thought the R-score was some sort of correlation should it not go between -1 and 1? Panel E, is the legend

the wrong way around? It's also not clear what is meant by the dendrogram. I also struggled to find details about how the clusters were chosen, I do apologies if I missed them and whether any robustness analysis of this clustering was performed?

Figure 5 b, again I'm unsure about the details of the dendrogram and clustering performed here. There is an unexplained connect between C, D and F – are they the same subcellular niches? I couldn't find how these contours were made and what they mean. They look similar to those in the manuscript of Crook et al. 2021 (<https://doi.org/10.1101/2021.01.04.425239>), did the authors borrow this approach, if so it should be referred to? Perhaps this would be better as a ternary plot anyway?

Did the authors perform any controls on the treated cells to confirm that the treatment was working?

I don't really follow the argument on page 7 when the authors divide their hits into 4 clusters and then argue for different insights for each of these groups. For example, I can't really see how the authors came to "all these proteins shifted toward the endosomal profile" – how can we see this exactly? Furthermore, the authors states "only one changed significantly in abundance". This seems to suggest the authors also collected other experimental proteomics data, but I could not find any details or expected data in the supplementary material.

I don't really see how the data can be produced in 2.5 hours per map. There are multiple hour incubation steps and overnight digests, I think the authors could be a little bit more honest here about how long this protocol takes.

How was peptide quantitation combined to generate protein/group quantities? Could the authors provide peptide level data?

In the principal component analysis section, it's unclear what the authors mean by (0-1) normalised, did they divide by the max intensity in each fraction? The authors also write "scaled to unit variance in each fraction", how exactly was this done?

The authors frequently mention "distance" but there are many distances, which one did they use and why is it appropriate?

How many markers were used per-class for the SVM, the SVM needs a minimum number of markers for the cross-validation to be performed and if the class imbalance is high then a weighted SVM needs to be used.

The authors write "optimized parameters were used to classify non-marker proteins applying leave-one-out validation". I can't see how this can be true. Once the SVM is trained on all the marker protein (with hyperparameters chosen using cross-validation), the SVM is simply applied to predict the classes – i.e. no cross-validation is needed. Could you the authors be clearer about what is happening here?

I'm quite confused by the protein movement analysis as I detail is several comments:

Firstly, why was the 1K fraction removed? Then what is this multi-dimensional outlier test. It is left unexplained. The authors produce p-values – what is the distribution being used here? Why is the median of p-values used – there are standard approaches for combining p-values (such as Fishers method). I also don't see why it necessary to then convert this p-value into a score, why not just use the values that come out of the outlier test. The workflow score-> p-value -> score appears to be an attempt to make the method statistically rigorous but is rather confusing.

The R-score is also confusing. Why not keep all scores in the combined analysis? Pearson assume bivariate normality – is this the case for these data?

Why is a combined score used? The reason for scaling to the power 7 seems quite unusual. Does the R-score always need to be take to some power that depends on the data? How are users of this approach meant to choose this power or do you always recommend 7? Also R-scores are less than 1 so R^7 would be very small $0.9^7 = 0.478$. Would it not be best to normalise such that R-score and M-score are on the same scale? I would suggest the MR-plot show R^7 vs M if that is what's actually going to be used. "R-scores are strongly weighted" – do the authors mean strongly down-weighted?

The FDR estimation is unusual for several reasons. It hard to follow because of the lack of textual precision and mathematical clarity. I find the following sentences confusing: 1) "not expected to yield relevant shifts", do the authors mean there should be no shifts? Saying expected means there could be shifts but that would be surprising, but I think that is not the case? 2) What does largely mean when the authors say "largely static"? 3) What does "stringent" mean in "stringent FDR"? 4) Why are suddenly different combinations of R, M, p-values, correlations suddenly used? It doesn't make sense to compare these different statistics like this. 5) "FDR was calculated as the proportion ..." – so the authors mean the false discovery proportion? 6) Many p-values were computed but I do not see any sign of multiple testing correction.

Beside the previous explanation, the FDR computations makes very little sense. Firstly, consider a statistic with a mock dataset. It is actually not possible to estimate an FDR from this alone because the false discovery rate is the rate of false discoveries amongst all the discoveries. FDR is also not defined for 0 discoveries. If in a mock dataset 1 discovery is made at some threshold of that statistic, then the FDR is 100% - all of the discoveries were false. The authors choose to combine their mock and actual statistic together to get around this, but this causes a new problem. Suppose there are 5000 statistics produced from each of the mock and actual datasets and there are 100 true discoveries. Let us pick a threshold which include all the proteins then the estimated FDR by the authors' approach is 50%, but the actual FDR is 98%. So I'm not convinced this approach can work. Finally, the authors make their decoy dataset from their within control and within treatment analysis. Again, this causes an issue because the variability will be different within these datasets and so the scale of the scores will change. Even if this method is to be retained, it needs to carefully reworked. The authors cannot claim that this approach gives them statistical "significance".

There are a couple of ways the authors could remedy this approach. Firstly, we can assume that the majority of the protein do not change their localisation. The authors can use their outlier statistic (M)(don't compute p-values) and their reproducibility statistic (R) for all their values. There is no need to take medians, second highest, etc. For each protein combine these 6-values into a single score using the geometric mean. If the assumption behind the previous analysis is correct then these scores will be high amongst those that are changing their localisation. If the authors don't need p-values then they can simply look at the top ranked proteins and as a suggestion to follow up with microscopy. If they must compute p-values, they can use the assumption that most proteins do not change localisation to their advantage and estimate a null distribution from the data. There is an array of methods that could be appropriate, such as permutation/randomisation methods, bootstrapping, and the local FDR. Another simple approach would be to look at the results from the SVM and see which proteins were classified differently in each condition. Note that if an FDR 5% threshold in each dataset, then the FDR for a change will be higher.

I don't really understand supplementary figure 4 b/c. I don't see the turquoise colour in the row clustering of B. The same colours are repeated throughout the columns clustering and the row

clustering which is confusing. How do you get GO terms associated with the fractions? What happens to the “various” class in C? I think there are perhaps some annotation mistakes which means I can’t currently interpret what’s happening in these figures.

QC Tool

This looks good and probably useful for the community, but I would appreciate knowing what other tools are out there and what they can each do. Could the authors be explicit about what their tools does that can’t be done previously?

On intra-map scatter I get the following error, by following the tutorial

```
Traceback (most recent call last): File "/opt/gitrepos/DOMQC/webapp/QCtool.py", line 969, in update_comp_bp_global multi_choice, clusters_for_ranking, min_members=min_n, reference = reference_map) File "/opt/pythonenvs/domqc/lib/python3.6/site-packages/domaps/domaps.py", line 2416, in plot_biological_precision width=200+100*len(multi_choice), template="simple_white", height=400, points="all") File "/opt/pythonenvs/domqc/lib/python3.6/site-packages/plotly/express/_chart_types.py", line 568, in box layout_patch=dict(boxmode=boxmode), File "/opt/pythonenvs/domqc/lib/python3.6/site-packages/plotly/express/_core.py", line 1861, in make_figure for val in sorted_group_values[m.grouper]: KeyError: 'Experiment'
```

The PCA plot looks odd – I can’t see the axis properly:

I get stuck on the SVM analysis part because I don’t use Perseus, could the authors provide the outputs so that the SVM part can be done?

In lot of places the axis overlap with labels, boxes overlap with text and I can’t read the numeric values of certain options. I think the app could be cleaned up a bit.

Is it possible to change the colour of the plots?

Reviewer #2 (Remarks to the Author):

Since the package is still “under development”, can the authors add a NEWS.md file to their repository to that bugs/updates can be tracked. It might be quite frustrating to use this tool at the moment if there are lots of frequent changes.

The authors should hold themselves to a high standard of software development. The authors should advertise their test coverage somewhere, incorporate continuous integration tools and probably submit their package to a public repository.

I’m not quite clear on the workflow of bioinformatics analysis for these data and where the tool fits into this workflow. Could the authors add a short section which summarise the major bioinformatic steps to process and analyse the data?

When trying the tool from another spatial proteomics dataset. I ran into a number of issues. This was 0/1 normalise quantitation data and I tried lots of different formats to get it to work. The error is somewhat unhelpful and the authors should make sure helpful errors are outputted:

Traceback (most recent call last): File “/opt/gitrepos/DOMQC/webapp/QCtool.py”, line 603, in execution i_class.run_pipeline(content=BytesIO(i_file.value), progressbar=analysis_status) File “/opt/pythonenvs/domqc/lib/python3.6/site-packages/domaps/domaps.py”, line 169, in

A grey box appears over the error message, which makes it difficult to read:

```
Traceback (most recent call last): File
"/opt/gitrepos/DOMQC/webapp/QCtool.py", line
603, in execution
i_class.run_pipeline(content=BytesIO(i_file.value),
progressbar=analysis_status) File
"/opt/pythonenvs/domqc/lib/python3.6/site-
packages/domaps/domaps.py", line 171, in
"/opt/pythonenvs/domqc/lib/python3.6/site-
packages/domaps/domaps.py", line 385, in
custom_indexing_and_normalization
[re.match(self.name_pattern, col).group("rep") for
col in df_original.columns] if not "" in
self.name_pattern File
"/opt/pythonenvs/domqc/lib/python3.6/site-
packages/domaps/domaps.py", line 385, in
[re.match(self.name_pattern, col).group("rep") for
col in df_original.columns] if not "" in
self.name_pattern IndexError: no such group
```

Reviewer #2 (Remarks to the Author):

There seem to be only 3 organisms supported which is somewhat disappointing for a general tool. Is the increased spectral complexity of DIA mean that it cannot be applied outside these organisms

Traceback (most recent call last): File "/opt/gitrepos/DOMQC/webapp/QCtool.py", line 603, in execution i_class.run_pipeline(content=BytesIO(i_file.value), progressbar=analysis_status) File "/opt/pythonenvs/domqc/lib/python3.6/site-packages/domaps/domaps.py", line 169, in

Reproducibility

I could not find code to reproduce the figures in the text – could the authors provide a workbook or equivalent so the analysis can be followed?

Minor comments:

The authors write “organelles” frequently, but some classes are not organelles.

The authors often use the term “shift profiles”, “partial shift”, “marginal shift” and “parallel shift” – I’m not sure what these terms are meant to mean biologically.

Supplementary table 2 is a database? I couldn’t find a link to a database anywhere. If it’s just a table I would call it that rather than a database as I was expecting something else here.

Point-by-point response

Before we address individual comments, we would like to thank all three Reviewers for their positive feedback and their constructive criticism. We would also like to summarize the major changes and improvements we have made:

1. We have substantially expanded the phenotypic characterization for our starvation/BafA experiment (**new main Figure 7**). Specifically, we performed a detailed time-course of BafA treatment to follow the movements of six Golgi proteins over eight hours by quantitative immunofluorescence microscopy. The results fully confirm the very high sensitivity and quantitative predictive power of our DOMs translocation analysis, and shed new light on the different types of cycling behaviour of Golgi proteins. Furthermore, we have extended our analysis of starvation induced translocations with new proteomic data (**new Supplementary Figure 4**), and augmented the interactive database of protein localization changes with new analysis functions, to facilitate the interpretation of individual protein translocations (**new Supplementary Table 3**).
2. Based on the comments of Reviewer 2, and on extensive feedback from other labs already using the QCtool for their data analysis, we have drastically improved the functionality and usability of the QCtool:
 - a. The tool is now much more flexible towards different use-cases, for example different organisms and custom input formats, and offers more control over the analysis workflow;
 - b. it reports several additional quality metrics, and allows easy integration of external data, such as the outputs from MetaMass and Perseus;
 - c. it is even more transparent and user-friendly, as it now features a system to save and load settings, improved documentation, and automatic software tests.
3. Since the MaxQuant DIA processing software evolves rapidly, we have re-analyzed all our proteomic data with the latest official release of MaxQuant (2.1.3), and updated all figures accordingly. The new processing shows an even more striking advantage of DIA over DDA for dynamic organellar maps.

Reviewer #1 (Remarks to the Author):

Schessner et al. describe a novel method to create Dynamic Organellar Maps (DOMs) in human cells by combining fractionation and DIA acquisition and claim a substantial improvement in coverage and timing. Although this is potentially interesting for the proteomics community, I do not see the novelty here, rather a much-improved version of their previous published method. Even in the introduction (starting from the second paragraph and until the end) the authors give very technical details of the previous method which are not necessarily useful for the broader readership of Nat Communications which also include experts in cell biology, biochemistry, cell signalling, cancer, etc. The first part of results is again very technical and regards the development of software for data analysis which would be more suitable for a specialistic proteomic journal. Also evaluating the performance of LC is more suitable for proteomics journal than for the broader readership of this journal.

Studying the effect of starvation and bafilomycin A1 treatment is interesting but the authors do not explain why they want to use these treatments (which is the big picture, the big biological problem to solve here?) or why they developed this method to study autophagy and changes in protein localization. However, the finding that “GLG1 showed complete transition from Golgi to a punctate endosomal pattern, TGOLN2 underwent a partial translocation, and GALNT2 retained its Golgi pattern (Fig. 5G)” is potentially very interesting. Any functional follow up? What happens whether this translocation does not happen? Validating this novel idea would make the paper novel.

>We would like to thank the Reviewer for their encouraging feedback. As the Reviewer points out, our new DIA-DOMS workflow is ‘a much-improved version’ of our previous method. A central aim of our study is to make phenotype discovery through dynamic organellar mapping as accessible as possible to the cell biology community. On the one hand, we leverage the latest advances in mass spectrometry data acquisition and software development, to make the method as effective as possible and, thus, useful even for labs with limited mass spec capacity. On the other hand, we develop a comprehensive and user-friendly data processing pipeline, via our QC Tool. To the best of our knowledge, the QC tool is unique in the field of spatial proteomics: usage does not require any programming skills, it has a self-explanatory graphical user-interface with extensive help functions, it accommodates a wide variety of data input formats, and offers a large range of data formatting, analysis and quality evaluation functions. All of these features are accessible online, and no software installation is required. We thus hope to lower the entry barrier for spatial proteomics experiments as much as possible, and to entice cell biologists to venture into the field.

The manuscript principally reports a methods development study, to provide a fully integrated high-performance organellar mapping pipeline. In our original submission, we aimed to keep the technical detail in the main text to a minimum, and off-loaded a large part of the technical details and method development into the supplemental material and Methods section. Furthermore, we included a conceptual introduction to the method and DIA mass spectrometry, which we feel is necessary to convey to non-specialist readers how the method works and why the new workflow performs so well. However, we appreciate the Reviewer’s concerns about the technical complexity and want to make the manuscript as accessible as possible to a wide readership. Therefore, we have completely restructured the introduction, which now more broadly introduces spatial proteomics (which was also requested by Reviewer 2). We have removed technical jargon wherever possible, and added two explanatory ‘boxes’ that contain a glossary of technical terms frequently used in the manuscript (Box 1), as well as brief descriptions of metrics used to evaluate spatial proteomics data (Box 2 – this was also a suggestion from Reviewer 3). With these changes, the manuscript will be even more accessible to non-specialist readers.

The main purpose of the starvation/BafA treatment experiment is to test the performance of our method in an important, but only partially explored, cell biological context, to provide both positive controls and discover novel biology. We picked the widely-used starvation/BafA treatment, since based on the literature we expected that it would induce a variety of subcellular localization changes. To our knowledge, these have never been studied systematically with spatial proteomics, making it an ideal test-case for our mapping approach. Our data provide a feature-rich phenotype discovery resource, which will be of interest to many cell biologists and especially to the large autophagy community. Readers can now investigate *in silico* if a protein responds to the treatment with a subcellular localization change, which would indicate that it may be mechanistically involved, or affected. We are not aware of the existence of

any similar resource. To further enhance its accessibility and usefulness, we have now substantially improved the functionality of our interactive supplementary table (**new Supplementary Table 3**).

Furthermore, we used the dataset to drill down on one specific effect of the treatment, the endosomal trapping of Golgi proteins, which occurs in response to BafA. Although the phenomenon itself was first reported over 20 years ago based on a candidate approach, it has never been studied systematically, so the question of which Golgi proteins cycle to endosomes had remained open. Here, we provide the first global evaluation of cycling and report that the Golgi contains two sets of proteins, cycling (e.g., GLG1/TGOLN2) and non-cycling (e.g., GALNT2). Furthermore, our quantitative mapping data predicted that there are marked differences in either cycling kinetics or cycling pools among the cycling proteins.

Following the Reviewer's suggestion, we have now further explored these differences, using an orthogonal method. We performed an eight-hour time-course experiment of BafA treatment and followed the localization changes of six different Golgi proteins by quantitative fluorescence microscopy (**new Figure 7**). The six proteins were chosen to represent different classes of cycling Golgi proteins, as predicted by our proteomic data: non-moving proteins, partially translocating proteins, and fully translocating proteins. Our new analysis reveals that Golgi proteins have highly individual cycling patterns, ranging from stationary at one end to fast complete transitions, via intermediate phenotypes. Importantly, these changes are in excellent agreement with the predictions made by our proteomic translocation analysis, supporting the sensitivity, specificity and quantitative nature of our approach.

The reviewer further suggested to investigate functional reasons and consequences of endosomal cycling for individual Golgi proteins. While we completely agree that these are important questions, functional investigations at single protein level are beyond the scope and aims of this systematic study. Nevertheless, we are convinced that our global identification of different classes of cycling Golgi proteins will greatly facilitate future investigations of the phenomenon.<

Reviewer #2 (Remarks to the Author):

The Borner lab present an updated version of the DOM workflow, combining it with DIA-MS. The manuscript suggests that their approach is more applicable and more straightforward to implement. I can see that this method makes their approach more widely accessible. Their updated workflow is accompanied by new software tool for quality control of their workflows. I commend them for making it open source as too much proteomics software is not and holding back the community. I get a positive impression from the manuscript and can see it as a valuable scholarly contribution. The authors attention to detail is very high, which makes the manuscript enjoyable to read.

>We would like to thank the Reviewer for their positive assessment of our manuscript and for highlighting particularly strong points.<

I remain to be convinced on several point in the manuscript, however, and I propose that the authors can address them to make their manuscript more solid (and in some parts mathematically correct). I hope the authors find my comments useful to improve their manuscript.

>We thank the Reviewer for this encouraging evaluation and for their very detailed and helpful comments to improve the manuscript. We have augmented, corrected and modified the manuscript accordingly.<

Overall comments:

The manuscript describes an approach that contains elements that are similar to already published and established methods. This would not be much of an issue provided that this work was more carefully positioned with respect to these methods in terms of comparisons and inclusive citations. I find some parts of the manuscript somewhat oversold and I would hope to see a more realistic description of what the method can offer in the final version of the manuscript.

In particular, I have two points that may require some significant re-thinking on the part of the authors that are more high-level concerns rather than points about specific arguments. The manuscript would benefit from addressing these points.

Firstly, the method is somewhat similar to the approach published earlier in the year by the Olsen/Lund-Johnson lab. I appreciate that publication uses chemical fractionation, but other aspects are quite similar. Furthermore, the depth of application and demonstration in that paper is considerably higher (phosphoproteomics and non-cell line samples) and from reading both manuscript methodological details, the Olsen/Lund-Johnson method seems to require less starting material. There is a very cursory comparison in the discussion, and I think it would be better if readers could see much more clearly the differences between the approaches. This could include MS-time, instrument/reagents needed, applicability, reproducibility, resolution. I think a more quantitative comparison is needed.

>We agree with the reviewer that there are some similarities between our method and the Olsen/Lund-Johansen method (Martinez-Val et al., Nature Comms, <https://www.nature.com/articles/s41467-021-27398-y>.) Both are global spatial proteomics methods based on mass spectrometry of a small number of cellular fractions. However, there are many quite pronounced differences, in operation, scope of application, and performance, as detailed below. Since it will be important for readers to appreciate these differences, to select the optimal spatial proteomics method for their purposes, we agree that a more detailed discussion is warranted. Following the Reviewer's suggestion, we have included an extensive quantitative comparison of method performance, as well as an overview of key differences and commonalities (**new Supplementary Figure 6**). Furthermore, we have now included a substantial discussion of the Olsen/Lund-Johansen Method in the Discussion section.<

Regarding the Reviewer's specific comments:

'Firstly, the method is somewhat similar to the approach published earlier in the year by the Olsen/Lund-Johnson lab.'

>First of all, we would respectfully like to point out that our DOMs approach was first published in 2016 (Itzhak et al., eLife), and has since featured in many other studies. Our method (albeit originally with SILAC DDA) thus pre-dates the Olsen/Lund Johansen Method by five years. The current study is an extension of our own work, and not of another lab's method.<

'I appreciate that publication uses chemical fractionation, but other aspects are quite similar.'

>The use of differential centrifugation vs chemical fractionation results in conceptually quite different approaches. While our method focuses on separating intact organelles, chemical fractionation disrupts organelles, which causes all luminal organellar proteins to appear in the same soluble fraction, and organellar membrane compartments to lose their distinguishing features. As a result, chemical fractionation has relatively low resolution of membranous organelles. DOMs, in contrast, specifically focus on resolving membranous organelles. To illustrate this point, we now include a comparison of compartment prediction accuracy, which shows the superior performance of DOMs for resolving membrane bound organelles (**new Supplementary Fig. 6**). Conversely, chemical fractionation offers higher resolution of sub-nuclear compartments, which we now clearly state in the discussion.

The Reviewer's comment about similarity probably refers to the fact that both methods use DIA, six subcellular fractions, and 21min Evosep gradients. However, in our study, the 21 min gradient format is only one of many formats that we investigate; unlike the Olsen/Lund-Johansen study, we predominantly investigate deep organellar mapping with long gradients. A main aim of our study is to provide a detailed analysis of the trade-off between investing more MS measurement time and the gain in resolution and depth. Our 100 min gradient maps have vastly better coverage, resolution, and reproducibility than the Olsen/Lund-Johansen maps. Such deep formats are not investigated in the other study.

Finally, the Olsen/Lund-Johansen study provides no software tools for spatial proteomics, whereas our study provides a complete analysis pipeline.<

'Furthermore, the depth of application and demonstration in that paper is considerably higher (phosphoproteomics and non-cell line samples) and from reading both manuscript methodological details, the Olsen/Lund-Johnson method seems to require less starting material.'

>We would like highlight that the DOMs method has been applied to a large number of cell types and tissues – published examples include HeLa (Itzhak et al., 2016), dendritic cells (Kozik et al., 2020, Cell Reports; Rodriguez-Silva et al., 2023, bioRxiv), and primary mouse neurons (Itzhak et al., 2017, Cell Reports). Since the actual map preparation part of the protocol is unchanged, we did not feel it necessary to include other cell types.

Regarding the starting material: The chemical fractionation in the Olsen/Lund Johansen study used a 15 cm dish of 70-80% confluent HeLa cells (i.e., probably around 10 million cells). While we used three times as much in our study, this was purely to obtain enough material to enable many different proteomic analyses from the same set of map samples (DDA, DIA, 21 min gradient, 44 min gradient, 100 min gradient, with and without triple fractionation, etc.). This was essential to ensure comparability by eliminating biological variation between experiments. However, it does not reflect the minimum starting material for our method. As detailed in our previous step-by-step protocol publication (<https://currentprotocols.onlinelibrary.wiley.com/doi/full/10.1002/cpcb.81>), a single 10 cm dish is the minimum recommended starting material for DOMs (i.e., half the starting material required by the Olsen/Lund-Johansen method). We have clarified this in the **new Supplementary Figure 6**.<

'There is a very cursory comparison in the discussion, and I think it would be better if

readers could see much more clearly the differences between the approaches. This could include MS-time, instrument/reagents needed, applicability, reproducibility, resolution. I think a more quantitative comparison is needed.'

>We agree with the Reviewer that a more substantial comparison would add to the manuscript. We have hence performed a detailed quantitative comparison with our 21 min format maps (**new Supplementary Figure 6**) and added corresponding statements to the discussion sections. We found that our method outperforms the chemical fractionation approach in most metrics, but also highlight that the two methods offer complementary features, in that DOMs have better resolution of membrane-bound organelles, whereas the Olsen/Lund-Johansen methods resolves sub-nuclear compartments.<

Secondly, the paper is somewhat a 3-1, a software tool, an improved DOM method and application to lysosomal pH disruption. The paper might be better split into three papers as this paper is quite difficult to follow what the contributions are. Currently, none of the 3 separate parts are that convincing for anything other than bioinformatics/proteomics journals. The software tool doesn't appear to offer anything other than simple metrics and things already implemented and easily useable elsewhere. Whilst there is a clear improvement by using DIA, it is not much of a conceptual advance to apply DIA to DOM and can be considered somewhat incremental. The authors suggestion of DIA SILAC DOMs or demonstrating that their approach finds new proteins in their lysosomal pH application would have greatly improved the manuscript. For example, it's not clear that their new-found proteins in their application could not have been found using their original protocol. It's not clear that just finding more proteins brings any benefit to the methodology.

>We take the Reviewer's point that the study needs to accommodate three seemingly separate parts – methods development, software development, and sample application. However, our intention is to provide an integrated and comprehensive analysis pipeline for spatial proteomics. The software development is hence intimately connected with the method development. The application to an important open question in cell biology is essential to test the usefulness of the approach. In our opinion, the 3-in-1 structure is thus required to develop the approach and demonstrate its power. With respect, we are convinced that all three parts provide major new contributions to the field (as pointed out by the Reviewer in their opening statement). Regarding the increased depth of our method – this is a major improvement over existing methods – we are unaware of any published spatial proteomics method that provides a 6000+ proteins depth in 12h of mass spec time. Since the aim of global spatial proteomics is to cover most of the proteome, good proteomic depth is essential, especially since less abundant proteins tend to be less well studied than the top few thousand 'housekeeping' proteins. Conversely, achieving the same coverage that has previously allowed us to make major new discoveries (Davies et al., 2018, 2022, Nature Comms) in <20% of the mass spec measurement time (3x21 min gradient now vs previously 3 x 120 min gradient) is a genuine performance leap.<

The application is somewhat under-developed and could be complemented by a number of assays to substantiate their claims about translation/autophagy changes. For example, I'm not at all convinced that ribosomal protein shifting towards to the highest speed fraction is consistent with "starvation-induced reduction of translational activity".

> Our interpretation of the shift in ribosome proteins towards the highest speed fraction is that this corresponds to a reduction in size of translational assemblies and/or increase in free ribosomes (i.e., the presence of ribosomal proteins in less dense structures). We would expect this change in distribution to

also be mirrored by an increase of ribosomal proteins in the cytosol. Following the Reviewer's suggestion, we have characterized the ribosomal shifts in more detail. We now include a full analysis of changes in the cytosolic fraction (**new Supplementary Figure 4F**). As we predicted, starvation/BafA treatment strongly and specifically increases the proportion of ribosomes and many translation-associated proteins in the cytosolic fraction, consistent with ribosomal inactivity/release from mRNA. Similarly, we have investigated nuclear transitions as part of the starvation response, and pick up known and novel phenotypes (**new Supplementary Fig 4E**).

Furthermore, we have now added a substantial new quantitative microscopy-based characterization of Golgi-endosome cycling behavior (**new Figure 7**), which is the focus of our biological application and provides new insights into Golgi protein cycling dynamics. Please see above (introduction and comments to Reviewer 1) for more details.<

Major comments:

The authors argue this is the simplest and easiest method to implement amongst the methods available. However, I couldn't really find any details about the other methods other than a selective set of cursory citations. Could the authors contextualise their method within the literature?

The same is true of their QC tool, could the authors contextualise this tool amongst those already available in the spatial proteomics community? What is offered beyond MetaMass and QSep?

We agree with the reviewer that a better embedding of our study in the literature will be helpful. We have therefore substantially augmented the Introduction and Discussion accordingly, and added the **new Supplementary Figure 6** for a detailed comparison with the Olsen/Lundberg-Johansen method.

Regarding the QCtool: There are indeed only few spatial proteomics software tools available, as most labs use their own custom data analysis pipelines. The only other intended end-to-end framework is pRoloc, developed by Laurent Gatto et al., which is written in R and includes QSep as a tool for assessing cluster separation. However, even with the companion library pRolocGUI, they provide little functionality in a graphical user interface, and it requires some degree of experience with both R and the underlying packages to be useful. Any quality visualizations have to be specifically generated by the user, rather than being provided immediately upon data analysis. Other tools, including MetaMass, only cover parts of the data analysis workflow. Thus, to the best of our knowledge (and as now highlighted in the Discussion), our QCtool is the first and only tool that provides all relevant data processing steps and visualized quality metrics, while requiring no programming skills and accommodating custom input data. We feel this is of critical importance to expand the use of spatial proteomics beyond specialized proteomics labs.<

The authors argue for improved reproducibility which I can mostly see from their analysis. The authors are missing simple x-y plots of the protein/peptide quantitation values at each centrifugation speed, where the data are compared one replicate against the other. Spearman correlation accompanying these would also add value. I can also see that a direct comparison to the Olsen data would be valuable here.

>We thank the reviewer for these suggestions. We have now included correlation plots in the QCtool. We also show a heatmap of sample correlations for our 100-minute gradient data in the **new Supplementary figure 2D**.

We now also include an in-depth comparison with the Olsen/Lund-Johansen Method in the manuscript.

As summarized in the requested method comparison (**new Supplementary Figure 6**) the Olsen/Lund-Johansen Method uses the same 21-minute gradient as we do as part of our gradient analysis, but used Spectronaut direct DIA for quantification. On the PRIDE repository they provide both the Spectronaut .sne file, as well as the protein group report file. To avoid quantifying biological differences as unwanted measurement noise, we reduced their report file to the 4 control condition replicates. We compared this to our own 21-minute dataset of 3 replicates and used Spectronaut rather than MaxQuant based results for maximum fairness. Since the Olsen output does not include any quality indicators, we used the same minimum 4-consecutive values filter in both cases.

The Olsen method quantifies profiles for 4541 protein groups in at least one of four maps and we quantify 4063 in at least one of three maps (**new Supplementary Figure 6A, B**), but with higher data completeness in our maps (ie more complete profiles). Thus, considering only profiles measured in multiple replicates and without relying on imputation, our DOMs provide significantly greater depth (3410 protein groups across the three DOMs vs only 2337 proteins groups across the three best Olsen maps).

We next assessed profile reproducibility using the functions in our QCTool (**new Supplementary Figure 6C**). The reproducibility across replicates was much better with DOMs than with chemical fractionation. Investigating further why this was the case and using our new sample correlation function suggested by the Reviewer, we found that one replicate in the Olsen/Lundberg-Johansen data is wildly different from the rest (**new Supplementary Figure 6D**). Removing this dataset increased the profile reproducibility to a range that is very similar as for our 21-min DOMs. Of note, this application of our reproducibility analysis highlights the necessity for quality control as provided by our tool, since the bad replicate was not recognized as problematic in the original Olsen/Lundberg-Johansen paper. For further comparisons, we thus evaluated the three best replicates from the Olsen/Lundberg-Johansen study against our three Dynamic Organellar Map replicates.

To gauge profiling precision, we calculated the intra-map scatter of stable protein complexes (**new Supplementary Figure 6E**). Dynamic Organellar Maps had substantially better overall precision, with the exception of nuclear complexes (Replication factor C, nuclear pore complex). This is consistent with the Olsen/Lundberg-Johansen method specifically mentioning a 'highly pure nuclear extract', and dynamic organellar maps specifically focusing on resolving organelles beyond the nucleus.

Running our pipeline for compartment predictions on both datasets, DOMs strength for resolving organelles becomes even more obvious, as the dynamic organellar maps reach an overall marker recall of 87% and the three best Olsen replicates only reach 74%. The differences are most pronounced for membrane organellar compartments (Lysosomes, ER, etc.), for which Dynamic Organellar Maps perform substantially better (**new Supplementary Figure 6F**).

The **new Supplementary Figure panel 6G** shows a table that highlights conceptual differences and similarities of both methods, as suggested by the Reviewer.<

Assessing the quality of the data through the F1-score is probably not the best idea. The issue is that it is highly susceptible to 1) the markers chosen 2) the strong class imbalance 3) machine learning methods chosen. The authors can remedy these points by considering the following analysis. The authors can change the marker set used, either by sub-sampling, considering different marker sets from the compartments database and/or using markers from different labs. The F1 scores are also presented without error bars, this can be improved by repeated sub-sampling of the hold-out test set used for prediction and comparing a distribution of scores. The strong class imbalance can be tackled with several analysis, a) being more explicit about the false positives and false negatives (I know the F1 score combines these but it's good to be explicit) b) show the distribution of scores over

the different subcellular niches and c) plot precision-recall curves and compute the area under the curve. Also demonstrating the results hold for say the k-NN algorithm and the random forest method would show that it's not simply over-optimisation to a particular ML algorithm but rather a generally improved methodology.

>We thank the reviewer for their comment and agree that any comparison of machine learning predictions has to be made with great care. For the purpose of benchmarking highly similar datasets it is crucial that the algorithm, marker set, parameters and evaluation method remain the same. As we made sure that this is the case, we believe all comparisons we show are fair. To ensure that our results are not dependent on the marker set and machine learning method, we also followed the workflow from the Olsen lab and evaluated our 100 min datasets using MetaMass II with the Thul predictions as markers. We include the plot here (see bar chart below), but not in the manuscript, as it ranks the experiments/methods in the same way as our SVM-based approach.

Following the Reviewer's suggestions, we have augmented the QCTool to include precision, recall, F1 score and class size, both as summary and as compartment-resolved plot. We already showed the F1 scores by organelle in a supplementary figure, but now use it in the main figure instead. Regarding the error bars on the F1 scores, we are limited by the output of Perseus. Perseus does not allow keeping a hold-out test set. Instead, we use 8-fold cross-validation of the full marker set for hyper parameter optimization to avoid overfitting, and then perform a leave-one-out cross-validation of the markers using these hyper parameters, which is how the misclassification matrix is generated. Perseus has no feature to consider class imbalances, but this does not appear to be a problem in our case, since the prediction performance for the smaller clusters is high (with leave-one-out cross-validation).<

I have some concerns with what the PCA plots are showing. Firstly, they are displayed incorrectly: they should show the variance explained in each component – otherwise they are uninterpretable, and the ratio of the axis should be comparable to the ratio of the variance explain. To be explicit the height of a PCA plot cannot ever be greater than the width.

>We thank the reviewer for their suggestions, based on which we have augmented the PCA pot functionality of the QCTool. We implemented a GUI-model for PCA plots that includes:

1. an elbow plot showing the variability explained by each principal component,
2. a loadings plot showing how each fraction contributes to the principal components, and

3. selection of 2D vs. 3D display, principal components displayed and highlighting of protein clusters.

These features are individually enabled depending on the context within the QCtool. Elbow and loading plots, and the cluster selection are displayed for global PCA plots. PCA plots showing individual protein clusters do not include these since they only show a slice of the global dataset.

Regarding the ratio of the axes, we agree that this would be an important aspect if the goal was to find clusters in the data de novo, as described in (<https://doi.org/10.1371/journal.pcbi.1006907>). However, this is not the case here. Our goal is solely to visualize the data in two to three dimensions to get a quick overview whether predefined clusters are separable, and whether the overall data structure matches the expectations of a successful experiment. Furthermore, please note that in general, the axis of PC2 can be longer than that of PC1, depending on the data structure – please refer to this review for some examples (eg Figure 2E and 3B show plots where PC2 has greater dimensions than PC1): <https://www.nature.com/articles/nmeth.4346.pdf?draft=collection> <

I'm also concerned that the authors suggest the best dimension to plot are 1 and 3. Strong experimental design would suggest that desired quantity (separation of subcellular compartments) dominates the variance in the data. However, the authors are suggesting that this is not true and there is some other reason for the scatter in PC2. Why is this the case? Is there a contamination driving the variability in this dimension? It would be good to investigate what is happening here.

>We thank the reviewer for this comment and we are happy to clarify our reasoning. We agree that a strong experimental design should lead to meaningful first principal components. That is indeed the case for our data, as shown by the loadings plots below. The first principal component is entirely driven by the fractionation gradient with the two highest speed spins (24K and 80K) loading it in the opposite direction as two early spins (3K and 6K) – see figure below. The 1K and 12K spins contribute little to this PC and consequently this is where most variability is left for PC2, with 1K alone pointing in the opposite direction to all other fractions. The reason we prefer showing PC3 instead is that our method's intended use is not to distinguish between organellar and nuclear proteins (which are almost exclusive to the 1K fraction), but rather to disentangle all other membrane bounded organelles. If we were to show PC2, half of the plotting area would largely be covered by nuclear proteins and the other organelles would be compressed strongly on the other half. PC3 on the other hand more strongly reflects the differences between the other organelles, as it is loaded almost alternately by the fractions along the gradient. Thereby it captures the more fine-grained profile differences beyond their peak fraction. Nevertheless, our QCtool can calculate and display as many PCs as there are variables, and users can freely choose the PC combination that optimally resolves their maps in 2D (or 3D).<

Finally, the Golgi proteins look quite different in each PCA plot (Fig. 2).

>We thank the reviewer for inspiring us to investigate this further. The figure below illustrates our findings. Firstly, the shape of the Golgi cluster changed slightly due to the reprocessing with MaxQuant V2. Now the two DIA analyses look much more similar to each other, and different from the DDA map. We picked out the protein MAN2A1 (circled in red) to illustrate to the reviewer why some Golgi proteins in the DDA maps fall away from the main cluster (see figure below). The profile plot shows that using DIA we are able to quantify the protein in the 24K and 80K fractions, but not with DDA. This can be explained by the low abundance of Golgi proteins in the last two fractions, paired with the stochastic nature of DDA, as outlined in the introduction of our manuscript. To ensure that this is the case here, we retrieved the peptide data from an 80K fraction. While DDA is able to identify almost as many peptides as DIA across the whole experiment, it was not able to quantify any of those peptides in the 80K fraction. Since both Mitochondria and the Golgi peak in the 3K fraction and are very low-abundant in the higher speed fractions, some Golgi proteins are mis-assigned to the mitochondria with DDA. The increased sensitivity of DIA greatly improves Golgi-Mitochondrial discrimination. In the manuscript, Supplementary Fig. 2C now also shows the improved sensitivity in several low abundant fractions.

To make this difference clear to readers we replaced the half sentence ‘but enhanced tightness of organellar clusters was apparent in the DIA maps’ from the Results sections with a more complete explanation:

‘Importantly, DIA maps showed fewer ‘outliers’ and reduced blending of compartment clusters (e.g., mitochondria and Golgi). Both effects are explained by the higher sensitivity of DIA, which leads to better quantification of proteins in fractions where they are of low abundance and, thus, yields more nuanced profiles (see also Supplementary Fig. 2C).’ <

I fail to find the nuclear compartment (it is just not labelled?)

>Indeed, there is no annotated nuclear compartment on our maps. As explained in the very first DOMs paper (Itzhak et al., 2016), we prefer to quantify nuclear and cytosolic pools of proteins separately from the organellar

mapping, to alleviate the multiple localization problem. Our DOMs workflow thus focuses on resolving the endomembrane system. The reason we still include the 1K fraction (which is heavily enriched in nuclear proteins) in our profile analysis is that it adds more information to some of the organellar profiles, thus helping for example to distinguish actin binding proteins from the Golgi apparatus. To analyze the nucleus and the cytosol, we employ a separate workflow, as originally described in Itzhak et al. 2016, analyzing the nucleus = 1K spin, and the cytosol = 80K supernatant, directly. This approach is particularly well suited to investigating soluble nuclear proteins, e.g. transcription factors and whether they reside in the nucleus or cytosol, as we demonstrate with the newly added analysis of these fractions in our comparative experiment (**new Supplementary Figure 4E, F**). If the user wishes to include a nuclear and or cytosolic compartment in the DOMs analysis, they can easily do it, as the QTool now allows upload of a custom marker set. Below is a PCA plot of our DDA dataset with markers from LOPIT-DC paper, which includes a nuclear and cytosolic compartment.

...and the DDA data looks like it has fewer outliers.

>As alluded to above in our explanation about the Golgi proteins, the opposite is now the case – DIA maps have fewer outliers. This is now pointed out in the main text. The reason is that the default settings for DIA processing in MaxQuant we used for our first submission were suboptimal. Specifically, the stabilization of large LFQ ratios is, according to the developers of MaxQuant, beneficial for DDA data, but detrimental for DIA data. That is why we now disabled this feature for the re-processing.<

I have to say I stared at Fig. 2E for quite some time to understand it and still do not really know what I'm expecting to see. I think it probably that the information density and colours that are making this plot difficult to read. Fig. 3C has the same problem. Is there a better way to display this information?

>We thank the reviewer for this feedback. What the figure was showing is the scatter of all protein complexes in all replicates – one such value per slice in the stacked bar graph. In DOM-QC we have now switched the default view for these data to a strip-plot, where we still show all the values, but as one distribution, rather than a stacked bar graph. By doing so the data are now a lot more intuitive to interpret, but a direct comparison of a specific complex is no longer possible in a static image. In the online tool we thus include an option to highlight all replicates of a specific complex, as well as retaining the stacked bar graph and the detail view of the data for a selected complex.<

It's hard to assess the scale of the differences in Fig. 2B. What do these changes in F1 score mean in practice and what is the associated error with these values?

>To make the practical impact of the F1 scores more obvious, we switched our Figure 2B with a version that shows all F1 scores per organelle. These scores are mostly important to gauge differences in quality of the predictions made using the SVMs trained on the same marker set. For example, Figure 2B tells us that Golgi predictions from the DDA dataset are not as good as with DIA. On the other hand, mitochondrial predictions are highly reliable in all three datasets.<

Figure 4 has some issues. Could the PCA plots be fixed as suggest above? I think panels B and C do not add much so could be sent to the supplementary materials. I am confused by panel D, I thought the R-score was some sort of correlation should it not go between -1 and 1? Panel E, is the legend the wrong way around? It's also not clear what is meant by the dendrogram. I also struggled to find details about how the clusters were chosen, I do apologies if I missed them and whether any robustness analysis of this clustering was performed?

>We thank the reviewer for these suggestions to improve Figure 4. As discussed above we use the PCA purely for gauging the distinction of previously defined cluster and therefore let the figure layout determine the scaling of the axes. We have moved panel B to the supplementary figure and removed panel C entirely. It is true that the R-score goes from -1 to 1, we now show this is Supplementary Figure 4B, but retain the zoomed version for the main figure. We have adjusted the colourscale of the heatmap to the more intuitive orientation.

Regarding the dendrogram and clustering, it is common in the field to perform hierarchical clustering analyses, which yields the similarity of both samples and proteins in the form of the dendrograms. For samples this is to ensure that replicate samples are closer than non-replicate samples. For proteins this serves as a means to put the proteins into groups based on their behavior. It is also common to manually chose the distance metric, linkage and distance threshold such, that any visually apparent groups are both complete and well separated at the same time. Afterwards the groups are usually 'labelled' by their behavior, or with the help of an enrichment analysis among all included proteins, the result of which is shown in Supplementary Figure 4C. We found that only complete linkage (as opposed to single or average) consistently yielded a similar dendrogram independent of the distance metric, so this is what we chose. We then chose the metric and cutoff such that a reasonable number of clusters became apparent with a large distance between all of them. We refrained from labelling the last cluster, because we could not get any clear enrichment. Furthermore, to enhance the clarity of the figure, we have removed the 'samples' dendrogram. The analysis workflow is now clearly described in the methods section.<

Figure 5 b, again I'm unsure about the details of the dendrogram and clustering performed here.

> The difference between the clustering in Figures 4 and 5 is that in Figure 5 we use the full pairwise correlation matrix as input for the hierarchical clustering. This way all pairwise distances are still visible through the heatmap and the relation between clusters can be interpreted. For the hierarchical clustering itself the same procedure as above was applied and the result is outlined in the methods section.<

There is an unexplained connect between C, D and F – are they the same subcellular niches? I couldn't find how these contours were made and what they mean. They look similar to those in the manuscript of Crook et al. 2021 (<https://doi.org/10.1101/2021.01.04.425239>), did the authors borrow this approach, if so it should be referred to? Perhaps this would be better as a ternary plot anyway?

>We thank the reviewer for pointing out the unexplained connection. The panels indeed show the same niches and we have now added a corresponding statement to the figure legend. The contours were created using a standard function in the plotting library used throughout the QcTool (https://plotly.com/python-api-reference/generated/plotly.express.density_contour.html), with default settings. While Crook et al. also use

this depiction, theirs is not the first use of such a plot; for example, we have used it in a previous spatial proteomics study as an overlay on the scatter PCA plot (Kozik et al., Cell Reports 2020); thus, in our opinion, no specific reference is required for this common method. We considered the option of using a ternary plot, but decided against it, as we'd rather conserve the undistorted positions in PCA space for depicting the movement of the proteins.<

Did the authors perform any controls on the treated cells to confirm that the treatment was working?

>We used immunofluorescence microscopy to confirm that the starvation and BafA treatments work as expected (Supplementary Figure 5C). We have now added a statement to the results section to clarify this point.<

I don't really follow the argument on page 7 when the authors divide their hits into 4 clusters and then argue for different insights for each of these groups. For example, I can't really see how the authors came to "all these proteins shifted toward the endosomal profile" – how can we see this exactly? Furthermore, the authors states "only one changed significantly in abundance". This seems to suggest the authors also collected other experimental proteomics data, but I could not find any details or expected data in the supplementary material.

>We thank the reviewer for their comment. To clarify how we came to these conclusions, we included **new Figure panels 4D-F** to exemplify the shift from Golgi to endosomes by three of our hits. The statement about the total protein abundance and nuclear and cytosolic pools stems from the separate analyses of the full proteome, 1K and supernatant samples. In the previous version we already included the full proteome analysis in supplementary Figure 4A, the other two analyses are new. All profile shifts are included in the interactive Supplementary Table 3, which generates figures akin to panel 4D-F.<

I don't really see how the data can be produced in 2.5 hours per map. There are multiple hour incubation steps and overnight digests, I think the authors could be a little bit more honest here about how long this protocol takes.

>The 2.5 hours refer only to the mass spectrometric analysis time, which typically is the resource bottleneck in spatial proteomics experiments. We never claimed that the whole procedure is that short. We apologize to the Reviewer that this was not explicitly clear from our description; it certainly was not our intention to mislead readers! We have checked the manuscript to prevent any such misunderstandings. Furthermore, the methods section (and several previous publications) provide a detailed account of how long the whole protocol takes.<

How was peptide quantitation combined to generate protein/group quantities? Could the authors provide peptide level data?

>Protein grouping is performed by the MaxQuant algorithm, and not part of our data analysis pipeline. While we agree with the reviewer that peptide level data would be an exciting extension of our method, this is beyond the scope of the current study. We provide peptide level data in our raw data repositories in case a reader is interested.<

In the principal component analysis section, it's unclear what the authors mean by (0-1) normalised, did they divide by the max intensity in each fraction? The authors also write "scaled to unit variance in each fraction", how exactly was this done?

>Each profile initially consists of up to six LFQ protein intensity values. We then sum these six intensities, and divide each intensity by the sum. As a result, the profile is now a normalized ‘percentage profile’, where all fractions sum to 1. All profiles thus become directly comparable, irrespective of the actual protein intensities.

For the PCA plots, we further center-scale the normalized intensities. For each fraction, the average is calculated, and subtracted from each intensity (centering). Subsequently, each intensity is divided by the intensity standard deviation of this fraction (scaling to unit variance). This is a common standard procedure for PCA.<

The authors frequently mention “distance” but there are many distances, which one did they use and why is it appropriate?

>Throughout the manuscript, we frequently use the Manhattan Distance (also known as ‘city block distance’, or absolute distance) as a distance measure. From past experience and previously published work (Itzhak et al., 2016, eLife; Matin-Jaular et al., 2021, EMBO J), we find that this metric performs particularly well for analyzing short 0-1 normalized profiling data. It is also a very simple and intuitive distance measure. In some instances, we use Pearson Correlation instead – this is more useful in cases where we look for profiles with similar trends/parallel features, but not necessarily identical features.<

How many markers were used per-class for the SVM, the SVM needs a minimum number of markers for the cross-validation to be performed and if the class imbalance is high then a weighted SVM needs to be used.

>We have added the number of markers used per class to Figure 2C. For all SVM analyses, we only include classes with at least 10 marker proteins. As explained above, Perseus has no feature to consider class imbalances, but this does not appear to be a problem in our case, since the prediction performance for the smaller clusters is high (with leave-one-out cross-validation).<

The authors write “optimized parameters were used to classify non-marker proteins applying leave-one-out validation”. I can’t see how this can be true. Once the SVM is trained on all the marker protein (with hyperparameters chosen using cross-validation), the SVM is simply applied to predict the classes – i.e. no cross-validation is needed. Could you the authors be clearer about what is happening here?

>The author is correct in their description of the first part of the workflow – we optimize the hyperparameters with cross-validation of the pre-defined marker set (leave out 1/8 of data). However, the optimized parameters do not specify a fixed position of the hyperplane that separates prediction classes; rather, they instruct the algorithm how the hyperplane should be fitted in the context of a given dataset. (SVMs maximize the separation of two pre-defined classes, by finding a hyperplane that separates them with the widest margin; this hyperplane depends on the data/class members fed into the algorithm; so the result will not just depend on the parameters used for the kernel function.) Perseus then removes a single marker protein from the set, applies the SVM algorithm with the optimized parameters, and checks if the held out marker is correctly predicted. This is then repeated for all markers in the set, one by one, and for each marker we then know if it was correctly predicted when it was not part of the set used to build the SVMs. This is effectively a second round of (leave-one-out) cross-validation. The correct and incorrect predictions are then counted for calculating the F1 scores.<

I'm quite confused by the protein movement analysis as I detail in several comments: Firstly, why was the 1K fraction removed? Then what is this multi-dimensional outlier test. It is left unexplained. The authors produce p-values – what is the distribution being used here? Why is the median of p-values used – there are standard approaches for combining p-values (such as Fisher's method). I also don't see why it is necessary to then convert this p-value into a score, why not just use the values that come out of the outlier test. The workflow score \rightarrow p-value \rightarrow score appears to be an attempt to make the method statistically rigorous but is rather confusing.

The R-score is also confusing. Why not keep all scores in the combined analysis? Pearson assumes bivariate normality – is this the case for these data?

Why is a combined score used? The reason for scaling to the power 7 seems quite unusual. Does the R-score always need to be taken to some power that depends on the data? How are users of this approach meant to choose this power or do you always recommend 7? Also R-scores are less than 1 so R^7 would be very small $0.9^7 = 0.478$. Would it not be best to normalise such that R-score and M-score are on the same scale? I would suggest the MR-plot show R^7 vs M if that is what's actually going to be used. "R-scores are strongly weighted" – do the authors mean strongly down-weighted?

The FDR estimation is unusual for several reasons. It is hard to follow because of the lack of textual precision and mathematical clarity. I find the following sentences confusing: 1) "not expected to yield relevant shifts", do the authors mean there should be no shifts? Saying expected means there could be shifts but that would be surprising, but I think that is not the case? 2) What does largely mean when the authors say "largely static"? 3) What does "stringent" mean in "stringent FDR"? 4) Why are suddenly different combinations of R, M, p-values, correlations suddenly used? It doesn't make sense to compare these different statistics like this. 5) "FDR was calculated as the proportion ..." – so the authors mean the false discovery proportion? 6) Many p-values were computed but I do not see any sign of multiple testing correction.

Besides the previous explanation, the FDR computations make very little sense. Firstly, consider a statistic with a mock dataset. It is actually not possible to estimate an FDR from this alone because the false discovery rate is the rate of false discoveries amongst all the discoveries. FDR is also not defined for 0 discoveries. If in a mock dataset 1 discovery is made at some threshold of that statistic, then the FDR is 100% - all of the discoveries were false. The authors choose to combine their mock and actual statistic together to get around this, but this causes a new problem. Suppose there are 5000 statistics produced from each of the mock and actual datasets and there are 100 true discoveries. Let us pick a threshold which includes all the proteins then the estimated FDR by the authors' approach is 50%, but the actual FDR is 98%. So I'm not convinced this approach can work. Finally, the authors make their decoy dataset from their within control and within treatment analysis. Again, this causes an issue because the variability will be different within these datasets and so the scale of the scores will change. Even if this method is to be retained, it needs to be carefully reworked. The authors cannot claim that this approach gives them statistical "significance".

There are a couple of ways the authors could remedy this approach. Firstly, we can assume that the majority of the proteins do not change their localisation. The authors can use their outlier statistic (M) (don't compute p-values) and their reproducibility statistic (R) for all their values. There is no need to take medians, second highest, etc. For each protein combine these 6-values into a single score using the geometric mean. If the assumption behind the previous analysis is correct then these scores will be high amongst those that are changing their localisation. If the authors don't need p-values then they can simply look at the top ranked proteins and as a suggestion to follow up with microscopy. If they must compute p-values, they can use the assumption that most proteins do not

change localisation to their advantage and estimate a null distribution from the data. There is an array of methods that could be appropriate, such as permutation/randomisation methods, bootstrapping, and the local FDR. Another simple approach would be to look at the results from the SVM and see which proteins were classified differently in each condition. Note that if an FDR 5% threshold in each dataset, then the FDR for a change will be higher.

>We would like to thank the Reviewer for their detailed comments and ideas for improving our outlier analysis. Our new workflow uses all six fractions (instead of only five, which was our previous standard approach), so this is now consistent with using six fractions throughout the manuscript. Importantly, we have taken up the Reviewer's excellent suggestions to apply the Fisher Method for combining p-values, and to omit the FDR control based on a mock dataset. Of note, the results are similar to our previous analysis, but we now follow more established mathematical procedures. Our new analysis workflow is much more straightforward, and also uses a much simpler FDR calculation. We now control the FDR for the M score only, using the Benjamin-Hochberg method, and use the R score as additional stringency filter.

Below is the description of the new workflow, as it is now reported in the Methods section:

'We compared organellar maps from untreated HeLa cells and from HeLa cells that had been starved in the presence of BafA for 1h. We used a modified version of our previously established MR analysis [13,44] to identify proteins with significant subcellular localization shifts. Replicates were numbered 1 to 3. Using the default settings of the QcTool, 5672 normalized high-quality profiles were obtained across all six maps. Next, 'delta profiles' were calculated within each cognate pair of untreated and treated maps, by subtracting the percentage profiles from a treated map from the profiles of its control map. For each of the three obtained sets of delta profiles, a multidimensional outlier test based on the robust Mahalanobis distance was performed [61], in the Perseus environment (v.1.6.2.3; proportion of data used = 0.75; 101 iterations). Obtained p-values reflect the probability of observing a given profile shift (or a greater shift) by chance. For each protein, the three p-values from the three replicates were then combined using the Fisher method. The combined p-values were corrected for multiple testing using the Benjamini-Hochberg False-discovery-rate (FDR) approach, and $-\log(10)$ transformed to obtain a Movement score (M score). We chose an FDR of 0.05 as our initial stringent cut-off, corresponding to an M-score of 1.3. As a further stringency filter, we also calculated a reproducibility score (R score). For each protein, the Pearson correlation of all pairs of delta profiles was calculated (Rep 1 vs 2, 1 vs 3, 2 vs 3) and the median correlation was designated as the R score. Profile shifts with $M > 1.3$ and $R > 0.75$ were considered significant and reproducible.'

I don't really understand supplementary figure 4 b/c. I don't see the turquoise colour in the row clustering of B. The same colours are repeated throughout the columns clustering and the row clustering which is confusing. How do you get GO terms associated with the fractions? What happens to the "various" class in C? I think there are perhaps some annotation mistakes which means I can't currently interpret what's happening in these figures.

>We thank the reviewer for the comment and apologize for the unclear labelling and unfortunate color choice. The sole purpose of Supplementary Figure 4B was to show the exact gene names per cluster. We have now removed the figure and instead included a **new Supplementary Table 2**, which shows the input data for the cluster analysis and the assigned clusters. To avoid confusion between the colors and make the heatmap more readable we removed the dendrogram for the fractions and show the fractions in their natural order. The terms displayed in the enrichment analysis in Supplementary Figure 4C are not the top-ranking terms per cluster, but

the ones below a Benjamini-Hochberg corrected p-value cutoff. This led to the label 'various', as there is no enrichment in this cluster making this cutoff.<

QC Tool:

This looks good and probably useful for the community, but I would appreciate knowing what other tools are out there and what they can each do. Could the authors be explicit about what their tools does that can't be done previously?

> We appreciate the reviewer's comment and happily clarify. There are no other software tools that combine the breadth of functionality and easy of use as DOM-QC. The only package that comes remotely close is the pRolocGUI library, which provides some visualizations using the pRoloc R package, but this is not useful to someone who does not have sufficient R programming skills to run such an analysis and launch the GUI. The only pure GUI that can in principle perform many of the analyses we include is Perseus, which however also requires significant expert knowledge and leaves the analysis and quality control strategy entirely to the user.

Conversely, we provide a graphical user interface for everything from data formatting, quality filtering, normalization, to visual and tabular outputs of data structure and quality. There is no need for knowledge of any programming language, as we host the tool online, free for everybody to use. This means that any biochemist can use the tool to establish their spatial proteomics method, using the benchmarking options to ensure a robust and optimal workflow, and analyze experiments, without having to involve a specialized bioinformatician. Since we collaborate with several other laboratories who want adopt our method, we already know that this is highly appreciated, and a huge timesaver even for spatial proteomics veterans.<

On intra-map scatter I get the following error, by following the tutorial

```
Traceback (most recent call last): File "/opt/gitrepos/DOMQC/webapp/QCtool.py", line 969, in
update_comp_bp_global multi_choice, clusters_for_ranking, min_members=min_n, reference =
reference_map) File "/opt/pythonenvs/domqc/lib/python3.6/site-packages/domaps/domaps.py", line
2416, in plot_biological_precision width=200+100*len(multi_choice), template="simple_white",
height=400, points="all")\ File "/opt/pythonenvs/domqc/lib/python3.6/sitepackages/
plotly/express/_chart_types.py", line 568, in box layout_patch=dict(boxmode=boxmode),
File "/opt/pythonenvs/domqc/lib/python3.6/site-packages/plotly/express/_core.py", line 1861, in
make_figure for val in sorted_group_values[m.grouper]: KeyError: 'Experiment'
```

>This error is now resolved.<

The PCA plot looks odd – I can't see the axis properly:

>We have addressed the issue by adding 5% to the default calculated range of the 3D PCA axes and zooming further out by default, so the axis labels should be visible from the start. Beyond this, the user can use the

interactive toolbar at the top of the plot to pan and zoom the plot. For static image export we always recommend the 2D projection.<

I get stuck on the SVM analysis part because I don't use Perseus, could the authors provide the outputs so that the SVM part can be done?

>We have included a **new Supplementary Table 4** containing all misclassification matrices generated using Perseus.<

In lot of places the axis overlap with labels, boxes overlap with text and I can't read the numeric values of certain options. I think the app could be cleaned up a bit.

>We thank the reviewer for their comment. We have made an effort to best calculate the size of all graphics based on the number of experiments/fractions/complexes displayed. This does not always work perfectly as the size of the legend depends on the length of the legend entries and thereby sometimes doesn't fit the margin of the plot and instead overlaps the actual graphic. We have found a way to avoid this problem and will adjust the code of all figures accordingly.<

Is it possible to change the colour of the plots?

>In order to change the color of the plots, one either needs to modify them in the downloaded svg, or using python. Generating the figures as in the jupyter notebook provided on github and with a dictionary of colors to assign to each legend item, one can use the following code (for scatter plots, similar for other modes):
figureobject.for_each_trace(lambda x: x.update(marker_color = color_dictionary[x.name]))<

Since the package is still "under development", can the authors add a NEWS.md file to their repository to that bugs/updates can be tracked. It might be quite frustrating to use this tool at the moment if there are lots of frequent changes.

>We thank the reviewer for their comment. We have already made an effort to develop the tool on github and with meaningful commit messages from the start and will make sure to formulate every bug we find as an issue prior to fixing it once we officially release version 1.0 at the date of final publication of the manuscript. We also included a NEWS.md file as suggested, where we outline major changes, and we will make use of the github release system from version 1.0 onwards.<

The authors should hold themselves to a high standard of software development. The authors should advertise their test coverage somewhere, incorporate continuous integration tools and probably submit their package to a public repository.

>We thank the reviewer for this valuable comment. The github repository now includes a github action that sets up an environment in different python versions and runs any doctests and pytest modules. We have included doctests for data transforming functions to provide unit tests and have written a pytest module, that simulates different user stories (also based on the files that were kindly sent to us through the editor), covering the analysis pipeline, as well as the new gui submodule of our package. We will also submit the project to pypi prior to final publication.<

I'm not quite clear on the workflow of bioinformatics analysis for these data and where the tool fits into this workflow. Could the authors add a short section which summarise the major bioinformatic steps to process and analyse the data?

>We thank the reviewer for providing this outside perspective on the transparency of our workflow. In brief, the workflow is to filter and normalize search engine outputs to obtain high quality profiles, which then serve as basis for differential analyses like the multivariate outlier test or subcellular localization predictions. Previously all of these steps had to be done manually in tools like Perseus and Excel, or with custom code e.g. in R or python. The tool fits into the workflow by standardizing the filtering and normalization steps, while also providing quality metrics. This saves the biochemist running an experiment a number of working days, depending on their level of bioinformatic experience, and makes the data analysis a lot more reproducible and transparent.<

When trying the tool from another spatial proteomics dataset. I ran into a number of issues. This was 0/1 normalise quantitation data and I tried lots of different formats to get it to work. The error is somewhat unhelpful and the authors should make sure helpful errors are outputted:

Traceback (most recent call last): File "/opt/gitrepos/DOMQC/webapp/QCtool.py", line 603, in execution i_class.run_pipeline(content=BytesIO(i_file.value), progressbar=analysis_status) File "/opt/pythonenvs/domqc/lib/python3.6/sitepackages/domaps/domaps.py", line 169, in

A grey box appears over the error message, which makes it difficult to read.

>We are very grateful to the reviewer for taking the time to try out the tool with different datasets and providing us not only with this feedback, but with the datasets as well. This has prompted us to do a major refactoring of our upload interface and underlying code for data formatting. Previously, we provided a very limited custom upload function and very hard-coded upload for selected data formats. Now all data are processed by the same highly customizable formatting pipeline and we provide default settings for standard formats in an expandable dictionary inside our codebase. All datasets that were submitted to us by the Reviewer are now part of our testing routine, to ensure that any users of our tool will have continued support for their input formats. During this refactoring we added several additional options, which were partly hard-coded and partly unavailable in the previous version:

- Relabelling, ordering and removal of fractions
- Simple column filters, to e.g. filter by a global q-value
- Sample normalization by sum or median
- Reversal of log transformation prior to processing
- Selection of additional columns to keep on the side throughout the analysis
- Customization of annotation sources (see next point)<

There seem to be only 3 organisms supported which is somewhat disappointing for a general tool. Is the increased spectral complexity of DIA mean that it cannot be applied outside these organisms

>We thank the reviewer for this comment and will gladly clarify. The four organisms we support are four major model organisms across different areas of biochemical research, and the ones for which we know of spatial proteomics studies. Furthermore, the annotations we provide are manually curated, rather than automatically generated. To make the tool more generally applicable, we have changed the configuration of the annotations to now accept custom input files for organelles, protein complexes and gene names, for any organism.<

Reproducibility

I could not find code to reproduce the figures in the text – could the authors provide a workbook or equivalent so the analysis can be followed?

>We have included the notebook used to reproduce the analysis and figure generation for Figure 2 in the github repository. We will also make sure to exemplify the use of any new functionality we add in the future in a similar manner.<

Minor comments:

The authors write “organelles” frequently, but some classes are not organelles.

>We thank the reviewer for this correction. We have replaced ‘organelle’ with ‘compartment’ where appropriate.<

The authors often use the term “shift profiles”, “partial shift”, “marginal shift” and “parallel shift” – I’m not sure what these terms are meant to mean biologically.

>We have replaced the word ‘shift’ with ‘translocation’ where appropriate, and also provide some definitions in the **new Box 1**.<

Supplementary table 2 is a database? I couldn’t find a link to a database anywhere. If it’s just a table I would call it that rather than a database as I was expecting something else here.

>Supplementary Table 3 (formerly Supplementary Table 2) is indeed just an Excel file, but it is interactive, and provides graphical and numerical outputs for user defined queries. Following the Reviewer’s suggestion, we now call it an ‘interactive table’.<

Reviewer #3 (Remarks to the Author):

Schessner et al benchmark DIA-MS approaches for DOM measurements, present a web implemented tool for QC analysis of DOM experiments and perform DOM to characterize Golgi proteins that cycle through endosomes. This study is relevant to the field of spatial proteomics and the DOM method has the potential to be broadly applied due to its simplicity. The described method optimizations provide significant advantages, and the web application will be of value for proteomic researchers interested in using DOM. The endosome-Golgi experiment is well performed, and the authors followed up and validated several hits.

>We would like to thank the Reviewer for the very positive assessment of our study.<

Major Comments:

1. For the MS method optimization and comparisons (Figure S1) the author should provide additional QC measures. What was the protein coverage, peptide/protein CVs, MS2 measurement points per peak and peptide measurement missingness for the different methods?

>We thank the reviewer for their comment and have expand the figure accordingly (**new Supplementary Figure 1**). We have now included the peptide depth and coverage, as well as the average number of points per peak. Regarding the CV, we find other measures more useful. Since we deal with profile data, we’d

rather know whether these are reproducible as gauged by the inter-map scatter. Additionally, we have now included sample pairwise correlation plots, showing sample reproducibility at the fraction level (**new Supplementary Figure 2D**). In the QCtool these can be switched to scatter plots to directly compare samples.<

2. The paper includes a vast number of non-trivial scores that need to be understood to interpret the figures and text. The scores are mostly described in the method sections. It would enhance the clarity of the manuscript if the authors could introduce the meaning of important scores when they first mention them in the text. For example: intra-complex scatter, inter-replica scatter, F1 score, R score, M score

>We would like to thank the reviewer for this helpful suggestion to improve the accessibility of the manuscript. We have now added two “boxes” that contain a glossary of acronyms and technical terms (**new Box 1**), and short explanations of key metrics used throughout the study (**new Box 2**).<

3. It would improve the usability of the QC web application if there were some description (eg figure legends) for the individual plots and how different outcomes should be interpreted.

>Following the Reviewer’s suggestion, we have updated the help functions in the QC tool.<

4. It is unclear how low, medium, and high-confidence localization prediction are determined/defined.

>The confidence classes are based on the machine learning SVM scores. We apologize to the reviewer for not stating the definitions more clearly. We previously used the same classification (Itzhak et al., 2017, Cell Reports). We have now added a corresponding explanation to the Methods section.<

5. Regarding the stacked intra-complex scatter plots. It would be good if the authors provide some additional analysis/explanations for those. For example, Figure 2E: As the authors state library DIA leads to the overall lowest combined intra-scatter. However, for several complexes (eg mitochondrial ribosomes, MCM complex) library DIA has noticeably higher intra-complex scatter. Is there an explanation for this? Maybe it has something to do with differences in protein coverage?

>Based on the Reviewer’s comments, the corresponding remarks by Reviewer 2, and the feedback from other users of the QC Tool, we have now replaced the stacked intra-complex scatter plot with a much more intuitive scatter plot (e.g., Figure 2E). Within the QC Tool, users can now choose between several plot options for displaying intra-complex scatter.

Furthermore, the reprocessing with MaxQuant V2 has further increased the precision of DIA. Overall, DIA has much better precision (ie lower intra-complex scatter) than DDA. However, as the reviewer points out, for a few complexes in the periphery of the maps (in particular, mitochondrial ribosomal complexes), DDA seemingly performs better. We speculate that this may be caused by more nuanced quantification with DIA across all fractions - with DDA, detection of mitochondrial proteins is largely confined to the first four fractions, whereas DIA also picks them up in all six fractions (see Supplementary Figure 2C). Six-datapoint quantifications will be inherently noisier than four-datapoint quantifications. Hence, the less complete profiles obtained with DDA appear to be less noisy than the more accurate DIA profiles. We would like to stress that this is a rare exception to the overall trend we see. <

6. Regarding their PCA plots (Fig 2, 4, 5). How much variance is typically captured by PC1, 2 and 3? Would

be good to report, so that users have a reference. Why do the authors always show PC3 instead of 2? What is contained in PC2? Fig 5 C-F: are these based on the combined PCA plots from Fig4? Why do they show PC2 here instead of PC3?

>This comment is almost identical to a point raised by Reviewer 2. We repeat part of our responses here:

We thank the reviewer for their suggestions, based on which we have augmented the PCA plot functionality of the QCTool. We implemented a GUI-model for PCA plots that includes:

1. an elbow plot showing the variability explained by each principal component,
2. a loadings plot showing how each fraction contributes to the principal components, and
3. selection of 2D vs. 3D display, principal components displayed and highlighting of protein clusters.

As you can see from the elbow plots, the first principal component is entirely driven by the fractionation gradient with the two highest speed spins (24K and 80K) loading it in the opposite direction as two early spins (3K and 6K). The 1K and 12K spins contribute little to this PC and consequently this is where most variability is left for PC2, with 1K alone pointing in the opposite direction to all other fractions. The reason we prefer showing PC3 instead is that our method's intended use is not to distinguish between organellar and nuclear proteins (which are almost exclusive to the 1K fraction), but rather to disentangle all other membrane bounded organelles. If we were to show PC2, half of the plotting area would largely be covered by nuclear proteins and the other organelles would be compressed strongly on the other half. PC3 on the other hand more strongly reflects the differences between the other organelles, as it is loaded almost alternately by the fractions along the gradient. Thereby it captures the more fine-grained profile differences beyond their peak fraction. Nevertheless, our QCTool can calculate and display as many PCs as there are variables, and users can freely choose the PC combination that optimally resolves their maps in 2D (or 3D).

The plots in Figure 5 are contour plots derived from the data of the non-treated condition used in Figure 4, as the Reviewer suspected. For consistency we have switched these to also show PC1 vs PC3 now.

7. It would be very useful if the web-tool included a way to identify differentially localized proteins and generate M-R plots. The described analysis to get to the differentially localized proteins and M-R plots is not very streamlined (eg including Perseus analysis) and might be a real bottleneck for people interested in using the DOM method. However, I understand if implementation of this analysis in the webtool is out of the scope for this manuscript.

>We agree with the Reviewer that including the MR analysis as part of the QCTool will be a very useful future addition. We apologize that it will be beyond the scope of the current submission.<

Minor Comments

8. PCA from Figure 4 vs Figure 2. Do I understand correctly that PCA in Figure 2 shows all, proteins whereas PCA in Figure 4 only shows marker proteins? It would be good to plot the PCA in Figure 4 the same way as Figure 2 so that comparisons between Figures 2 and 4 are possible.

>To ensure comparability between these plots, we have run a PCA on all five datasets and now include the non-marker proteins in Figure 4 as well.<

9. Unclear meaning of: “enhanced tightness of organellar clusters was apparent in the DIA maps” – page 4.

>We have removed this unclear formulation, and expanded the explanation as follows:

‘Importantly, DIA maps showed fewer ‘outliers’ and reduced blending of compartment clusters (e.g., mitochondria and Golgi). Both effects are explained by the higher sensitivity of DIA, which leads to better quantification of proteins in fractions where they are of low abundance and, thus, yields more nuanced profiles (see also Supplementary Fig. 2C).’

10. Supplementary Fig. 2D is not linked or described anywhere.

>This figure is now referenced in the text.

11. FDR = 5% should not be described as “high stringency cut-off” – page 7.

>We have revised our outlier analysis, and removed this statement.<

12. Use “m/z” instead of “Th” (method section).

We have made this correction.

REVIEWER COMMENTS

Reviewer #1 (Remarks to the Author):

The authors have tried to address my comments and the revised version shows several improvements and clarify many points. Thank you for this. However, I still think that this is a manuscript for a more specialistic proteomic journal as the conceptual advancement in cell biology is kept to a minimum and basically this is an improved and very user-friendly version of their previous work.

Reviewer #2 (Remarks to the Author):

The manuscript `Deep and fast label-free Dynamic Organeller Mapping` is much improved over the original version. I have much fewer concerns over the technical parts of the manuscript and using the QC webtool is much more straightforward. The quantitative analysis and more detailed comparison with the Lund-Johnson approach is well-received. Whilst I am convinced the DIA version is better than the original DDA method, I am still confused by the central messages.

General comments about overall impression:

QC tool:

Whilst useful, I'm not sure what barriers this tool is actually removing. For example, I would still need to write some code to check numbers of PSMs per Protein Group, the reproducibility/quality of the data at this level, normalization, machine learning analysis, movement analysis. The QC tool provides an easy to use implementation that isn't methodologically advanced. The authors mention other tools for quality control but these are not compared to a meaningful manner.

DIA and short LC:

This is a technical advance for spatial proteomics and will make it more applicable but is more of a technical note for a proteomics journal.

Application:

Whilst this is a convincing application, it is somewhat disconnected from the central message of the paper - that the proposed method really is better. The application would have been much better received with a side-by-side comparison with the DDA-DOM method. In that way, we could see if these "movers" could only have been observed using the DIA-DOM method. At the moment, I'm not sure if translocation analysis - the central reason for using DOM would be better using the DIA-DOM method or the DDA-DOM approach. If I was to get more confident mover results from DIA-DOM than DDA-DOM then that would be an important advance.

Technical comments:

The referenced nature methods article about PCA plots is incorrect and it's easy to see the issue from the highlighted *PLoS Computational Biology* paper. To understand this: eigenvalues must be ordered in the PCA algorithm and so the first PC corresponds to the largest variance the second the second largest and so on. It follows that the PC2 cannot be bigger than PC1, the issue in the nature methods paper is widespread and distorts the visualization. For faithful PCA plots the aspect ratios should be meaningful as mentioned before. The authors plot contours, arrows and make comments about the scatter - these interpretations are only meaningful if these plots are made correctly.

I find the M-score difficult to read on the plots - would this not be better on the log scale?

Whilst the authors have answered some of my questions in the rebuttal, the clarifications haven't been transferred over to the manuscript. There include:

- 1) Scaling of the variance (this can be done many ways and is ambiguous as written)
- 2) 0-1 normalisation (again there are many ways to do this)
- 3) Distance (it's used ambiguously on page 23 before being used on page 24)
- 4) Average is used interchangeably to be harmonic mean and arithmetic mean (and perhaps sometimes also median but I cannot tell from the text.)

The comment regarding “optimized parameters were used to classify non-marker proteins applying leave-one-out cross-validation”. The comment about optimizing hyperparameters is well understood and then a second round of cross-validation using optimized hyperparameters also. There are still two issues:

1) The evaluation of the F1 score uses the same data for optimisation as for evaluation (albeit in different cross-validation splits), which is data leakage - a common mistake. The authors need to do this more carefully: first to remove the markers that are going to be used for evaluation of the F1 score (usually 20% or so). Then using cross-validation on the remaining 80% the authors can optimize the parameters.

2) For specifically non-marker proteins what is happening - cross-validation cannot be used here as you don't have any compartment annotation.

Reviewer #3 (Remarks to the Author):

The authors have addressed all my comments. The manuscript as well as the web resource have significantly improved. I particularly welcome that the web resource now accepts different file formats.

Point-by-point response to the Reviewers' comments

General remarks to all Reviewers: Overview of major improvements

We would like to thank all Reviewers for their helpful feedback. We have corrected and augmented the manuscript accordingly. Of particular note, we have substantially improved the functionality of our analysis suite DOM-QC, which we have renamed as DOM-ABC (Analysis, Benchmarking, and quality Control). DOM-ABC contains new modules for machine learning-based organellar predictions, as well as for the translocation analysis, which had both been performed with external software in our previous submission. DOM-ABC now provides a seamless end-to-end pipeline for the analysis and exploration of spatial proteomics data. We have re-analyzed all maps with the new functions of DOM-ABC. The results are similar, and do not change our main conclusions; however, since DOM-ABC has several enhanced features not available in the external software we used previously (such as error bars on F1 scores, rigorous splitting of the compartment markers into test and validation sets, automated grid search for SVM training, better quality filtering of profiles for MR translocation analysis), the new data are even more informative. For example, we now detect 164 translocating proteins in the comparative experiment (instead of 108 previously). We have updated the text and all figures accordingly.

Furthermore, following the reviewers' suggestion, we have performed a detailed side-by-side comparison of DIA vs DDA organellar mapping for the analysis of the starvation/BafA experiment (new Supplementary Figures 5 and 7). This demonstrates that DIA-DOMs provide biological insights that are not obtained with DDA-DOMs, and thus offer a superior method for comparative organellar mapping.

Below are our detailed responses to individual comments.

Reviewer #1 (Remarks to the Author):

The authors have tried to address my comments and the revised version shows several improvements and clarify many points. Thank you for this. However, I still think that this is a manuscript for a more specialized proteomic journal as the conceptual advancement in cell biology is kept to a minimum and basically this is an improved and very user-friendly version of their previous work.

>We thank the reviewer for their positive assessment of our previous revisions. As the reviewer points out, the manuscript is now much more accessible, and our method is very user-friendly. Indeed, our central intention is to give cell biologists without specialist proteomics expertise access to a seamless, cutting-edge spatial proteomics workflow. Importantly, we have further augmented the DOM-QC tool, which now includes and automates all aspects of the data analysis (including SVM machine learning-based localization predictions, and translocation mapping by Movement/Reproducibility analysis). To encapsulate the greatly expanded capabilities of our tool, we have renamed it as 'DOM-ABC' (**A**nalysis, **B**enchmarking, and **Q**uality **C**ontrol). Our study thus provides comprehensive and user-friendly end-to-end solutions for both the proteomics and data analysis aspects of the DOMs workflow. This is unprecedented in the field, lowers the entry barrier, and will allow spatial proteomics to become much more mainstream

and standardized. Therefore, we are convinced that publication in a journal with a broad readership of cell biologists is the most suitable platform for our work.

Furthermore, we respectfully disagree with the reviewer regarding the importance of the cell biological advance we present. Starvation in combination with BafA treatment is a common method to induce autophagy, and has been used in hundreds of studies to date. We provide the first systematic and quantitative analysis of the subcellular rearrangements induced by this treatment. Our database of localization changes is unique and will provide an important resource for the large autophagy community. Importantly, we also provide the first systematic investigation of endosomal trapping caused by BafA treatment, which has previously only been observed for a small number of proteins, studied on an individual basis. Our detailed proteomics and imaging-based analysis of cycling Golgi proteins and their kinetics not only demonstrates the power and quantitative nature of the DIA-DOMs approach, but also addresses some long-standing questions in basic Golgi biology.

Finally, as proposed by Reviewer 2, we have performed a side-by-side comparison of DDA-DOMs vs DIA-DOMs for the starvation/BafA experiment (new Supplementary Figures 5 and 7). This demonstrates that DIA-DOMs provide phenotypic insights that are not captured by DDA-DOMs.<

Reviewer #2 (Remarks to the Author):

The manuscript `Deep and fast label-free Dynamic Organeller Mapping` is much improved over the original version. I have much fewer concerns over the technical parts of the manuscript and using the QC webtool is much more straightforward. The quantitative analysis and more detailed comparison with the Lund-Johnson approach is well-received.

>We would like to thank the Reviewer for their very positive feedback on our revisions.<

Whilst I am convinced the DIA version is better than the original DDA method, I am still confused by the central messages.

General comments about overall impression:

QC tool:

Whilst useful, I'm not sure what barriers this tool is actually removing. For example, I would still need to write some code to check numbers of PSMs per Protein Group, the reproducibility/quality of the data at this level, normalization, machine learning analysis, movement analysis. The QC tool provides an easy to use implementation that isn't methodologically advanced. The authors mention other tools for quality control but these are not compared to a meaningful manner.

>We thank the reviewer for their comment and want to clarify which barriers are removed by our tool, and present the measures we have taken to improve the tool and the manuscript even further.

First and foremost, we have added the MR movement/reproducibility analysis and SVM-based subcellular localization predictions to the tool, which now provides a complete end-to-end analysis suite for comparative organellar proteomics. We therefore re-named the tool from 'DOM-QC' to 'DOM-ABC', as it now enables comprehensive **A**nalysis, **B**enchmarking and quality **C**ontrol in an accessible yet highly interactive and customizable format. The minimum required user input is the upload of a protein groups file – the entire analysis workflow can then be performed at the click of a few buttons, and delivers publication-ready figures. In essence, we have greatly lowered the entry barrier for non-specialists, yet at the same time provide an unparalleled seamless analysis pipeline.

While the reviewer is clearly a proteomics expert, our primary motivation is to open the field of spatial proteomics to cell biologists, as we believe this will lead to important advances in the understanding of many cellular processes. Peptide-level analysis goes far beyond the standard workflow we aim to disseminate here. While we plan to incorporate user-friendly peptide-level data exploration in future versions of DOM-ABC, we regret that it is outside the scope and remit of the present study. However, we have already made the code partially compatible with peptide-level data, to accommodate future developments. Furthermore, to make the statistics shown in Supplementary Figure 1 more accessible, we have included the code for these in the notebooks folder of our github repository.

While there are some other tools that perform individual aspects of quality control or analysis, none of them are as comprehensive as DOM-ABC; in addition, they require programming skills (e.g., pRoloc), offer limited customizability (MetaMass), or require expert knowledge to decide which quality metrics to look at (Perseus). Furthermore, several of the quality metrics (reproducibility and precision scoring) are unique to DOM-ABC, as is the MR analysis. In our discussion, we cite the other tools and highlight differences; following the reviewer's suggestions, we have now added further details about how DOM-ABC substantially improves on existing applications whilst offering unique user-friendliness and flexibility. We are convinced that DOM-ABC will become the gold standard analysis tool for spatial proteomics data.<

DIA and short LC:

This is a technical advance for spatial proteomics and will make it more applicable but is more of a technical note for a proteomics journal.

>As we have described in our response to Reviewer 1, we think that this technical advance needs to be heard by cell biologists to encourage them to use spatial proteomics. As we know from our many interactions with other laboratories, the adoption of DIA methodology is daunting for non-specialist labs, especially for a complex application such as organellar mapping. Again, our intention is to provide a detailed and rigorous establishment, optimization, benchmarking and application of DIA mapping, to make this superior method easily available to as many non-specialist labs as possible.<

Application:

Whilst this is a convincing application, it somewhat disconnected from the central message of the paper - that the proposed method really is better. The application would have been much better received with a side-by-side comparison with the DDA-DOM method. In that way, we could see if these “movers” could only have been observed using the DIA-DOM method. At the moment, I’m not sure if translocation analysis - the central reason for using DOM would be better using the DIA-DOM method or the DDA-DOM approach. If I was to get more confident mover results from DIA-DOM than DDA-DOM then that would be an important advance.

>We thank the reviewer for agreeing that our application of DIA-DOMs to characterize the effects of starvation/BafA is convincing. We are also very grateful for their suggestion to compare the performance of DIA-DOMs and DDA-DOMs in the context of this experiment. We have repeated all mapping experiments in Figures 4 and 5 with DDA, using the same samples and instrument set-up. The results are shown in the new supplemental Figures 5 and 7. As expected, DIA-DOMs consistently outperform DDA-DOMs. DIA-DOMs provide a much greater number of profiled proteins (4,475 vs 2,055), a substantial increase in the number of detected significant translocations (164 vs 128), and achieve better M- and R scores. As a result, DIA-DOMs provide a much richer characterization of the starvation/BafA phenotype. For example, DIA-DOMs detect an additional group of transcription factors that shuttle out of the nucleus in response to the treatment; DDA-DOMs did not pick these up. DIA-DOMs identify 15 cycling Golgi proteins, and 29 static ones; in contrast, DDA-DOMs identify only 8 cycling and 12 static ones. Hence, the long-standing question which Golgi proteins cycle vs which ones are static is much more comprehensively answered with DIA-DOMs. DIA-DOMs are much better for comparative organellar profiling than DDA-DOMs.<

Technical comments:

The referenced nature methods article about PCA plots is incorrect and it’s easy to see the issue from the highlighted plos computational biology paper. To understand this: eigenvalues must be ordered in the PCA algorithm and so the first PC corresponds to the largest variance the second the second largest and so on. It follows that the PC2 cannot be bigger than PC1, the issue in the nature methods paper is widespread and distorts the visualization. For faithful PCA plots the aspect ratios should be meaningful as mentioned before. The authors plot contours, arrows and make comments about the scatter - these interpretations are only meaningful if these plots are made correctly.

>We thank the reviewer for this insight. Following their recommendation, we have now included an additional option in DOM-ABC to scale the axes of the PCA plots by variance. As the PCA plots in Figures 2 and 4 are used only for a visual overview, we have left them as they were to maximize the visibility of our predefined organelle clusters. However, we have made sure not to overinterpret them in the text, based on viewing the variance-scaled plots within DOM-ABC. For the PCA plots in Figure 5, in which we visually compare the direction and magnitude of shifts, we applied the variance-scaling as recommended by the reviewer, and switched to displaying PC2 instead of PC3; neither of these changes affected our qualitative conclusions.<

I find the M-score difficult to read on the plots - would this not be better on the log scale?

>The M-score is already on a log₁₀ scale – please refer to the methods for a detailed description.<

Whilst the authors have answered some of my questions in the rebuttal, the clarifications haven't been transferred over to the manuscript. There include:

- 1) Scaling of the variance (this can be done many ways and is ambiguous as written)
- 2) 0-1 normalisation (again there are many ways to do this)
- 3) Distance (it's used ambiguously on page 23 before being used on page 24)
- 4) Average is used interchangeably to be harmonic mean and arithmetic mean (and perhaps sometimes also median but I cannot tell from the text.)

>We apologize for the incomplete update of our manuscript We have corrected all remaining issues according to the reviewer's suggestions.<

The comment regarding “optimized parameters were used to classify non-marker proteins applying leave-one-out cross-validation”. The comment about optimizing hyperparameters is well understood and then a second round of cross-validation using optimize hyperparameters also. There are still two issues:

- 1) The evaluation of the F1 score uses the same data for optimisation as for evaluation (albeit in different cross-validation splits), which is data leakage - a common mistake. The authors need to do this more carefully: first to remove the markers that are going to be used for evaluation of the F1 score (usually 20% or so). Then using cross-validation on the remaining 80% the authors can optimize the parameters.

>We thank the reviewer for their suggestions. We have addressed these points in the new implementation of the SVM module in our tool. Users can now specify the size of a hold-out test set, which is not used for training or validating the hyperparameters, but only for the final evaluation of F1 scores, as the reviewer recommended. Throughout the manuscript, we used 30% of the markers as a hold-out test set. The remaining 70% of the markers were used for optimizing the hyperparameters, with 5-fold cross-validation. F1 scores were then calculated from the hold-out dataset only.

Furthermore, we have used this opportunity to implement error bars for the F1 scores, using the solution the reviewer suggested in the previous round of comments. To assess the variability of F1 scores the test set is sub-sampled 10 times (class stratified 75%) and the mean scores +- standard deviation are reported. We have added error bars to all plots showing F1 scores throughout the manuscript.<

- 2) For specifically non-marker proteins what is happening - cross-validation cannot be used here as you don't have any compartment annotation.

>Since we no longer use Perseus for SVMs, the question is largely redundant. Nevertheless, we can explain

how our new algorithm predicts localizations. First, markers are split into a training set and a test set. The hyperparameters are then optimized with five-fold cross-validation and an automated grid-search, using the training set only. Next, training set markers are again split into 5 groups, each with 80% of the markers. SVMs are fitted to each marker set, and applied to non-marker proteins and test set marker proteins, to generate five sets of localization scores. These are then merged into model fitting probabilities that range from 0 to 1 for each compartment using Platt scaling, as described further in the documentation of the scikit-learn python library (<https://scikit-learn.org/stable/modules/svm.html#scores-and-probabilities>). Each protein is assigned to the highest scoring compartment. Prediction accuracy is then gauged by scoring the proportion of correctly assigned test-set markers.<

Reviewer #3 (Remarks to the Author):

The authors have adressed all my comments. The manuscript as well as the web resource have significantly improved. I particularly welcome that the web recourse now accepts different file fdormats.

>We thank the reviewer for their positive evaluation and for their constructive feedback that helped us to improve our mansucirpt.<

REVIEWERS' COMMENTS

Reviewer #2 (Remarks to the Author):

This revision is very welcome. The additional experiments have convinced me that these changes really do make a difference for DOMs and that these protocols will be easier to implement spatial proteomics whilst also generating higher quality data. This will enable many more insights from these data and also make it easier to extend the methodology for example to spatio-temporal analysis. The changes to the computational analysis are necessary to make the pipeline accessible to the target audience. I am now convinced that this work is a greater contribution than a simple set of technical advances. I appreciate the rigour and efforts that have gone into producing a robust method and manuscript.

One cosmetic comment is that the use of white space on the MR-plots is still not ideal with a lot of the information compressed close to the y-axis. I appreciate the M-score is already on the log scale perhaps an additional square-root transform would help? I leave it to the authors' discretion of whether this would be useful.

Oliver Crook (University of Oxford)

Response to the Reviewer's final comments

Reviewer #2 (Remarks to the Author):

This revision is very welcome. The additional experiments have convinced me that these changes really do make a difference for DOMs and that these protocols will be easier to implement spatial proteomics whilst also generating higher quality data. This will enable many more insights from these data and also make it easier to extend the methodology for example to spatio-temporal analysis. The changes to the computational analysis are necessary to make the pipeline accessible to the target audience. I am now convinced that this work is a greater contribution than a simple set of technical advances. I appreciate the rigour and efforts that have gone into producing a robust method and manuscript.

>We thank the reviewer for their positive evaluation of our study, and their very constructive feedback throughout the review process.<

One cosmetic comment is that the use of white space on the MR-plots is still not ideal with a lot of the information is compressed close to the y-axis. I appreciate the M-score is already on the log scale perhaps an additional square-root transform would help? I leave it to the authors discretion of whether this would be useful.

Oliver Crook (University of Oxford)

>We thank the reviewer for this helpful suggestion. We plan to implement an option for M-score transformation in a future update of DOM-ABC.<